🔓 | **Open Peer Review** | Environmental Microbiology | Research Article

# Broad-spectrum applications of plant growth-promoting rhizobacteria (PGPR) across diverse crops and intricate planting systems

Shixiang Zhang,[1] Fengmin Li,[2] Luyin Chang,[2] Qiyu Zhang,[2] Zhenyu Zhang,[2] Tianci Wu,[2] Yanlin Wu,[3] Zhi Wang,[3] Yulong Su,[4] Xingyou Yang,[5] Weichang Gao,[6] Mengsha Li,[7] Yue Wu,[8] Ying Jiang[2]

**ABSTRACT**  Multifunctional plant growth-promoting rhizobacteria (PGPR) have garnered significant attention in agricultural applications; however, a few have applied them in crop rotation or intercropping fields. To identify PGPR with strong colonization ability and broad spectrum benefit, we screened strains from the local tobacco rhizosphere and evaluated their growth-promoting effects across various crops and farming systems. In this study, strain L8, identified as *Bacillus thuringiensis*, was selected as a multifunctional PGPR capable of producing indole-3-acetic acid (IAA), solubilizing potassium, and mobilizing both organic and inorganic phosphorus. Compared with the control group, the soil treated with strain L8 in tobacco pot experiments showed significantly higher levels of IAA, available potassium, and available phosphorus. Furthermore, tobacco plants inoculated with L8 exhibited significantly improved growth parameters compared with the uninoculated control. The results indicated that compared with the control, strain L8 increased tobacco fresh weight (by 83.18%), plant height (by 29.32%), relative chlorophyll content (by 14.33%), as well as plant phosphorus (by 23.78%) and potassium (by 30.81%). Interestingly, L8 not only enhanced tobacco growth but also improved tobacco root morphology, significantly increasing root length (1.55-fold), root surface area (1.78-fold), and root volume (2.05-fold) compared with the control. These findings provide valuable insights into utilizing plant growth-promoting bacteria for tobacco production. Moreover, the inoculation strain L8 enriched soil organic matter and nutrient content in both wheat (*Triticum aestivum*)-maize (*Zea mays*) rotation and peanut (*Arachis hypogaea*)-maize intercropping systems. Furthermore, within the intricate tillage systems of wheat-corn rotation and peanut-corn intercropping, multifunctional PGPR strain L8 can play a crucial role in crop growth promotion, yield increase, and higher soil nutrient availability.

**IMPORTANCE**  These findings aim to study the application of plant growth-promoting rhizobacteria (PGPR) in diverse crops and complex planting systems, showing their adaptability and effectiveness in various agricultural contexts. The study suggests that this strain improves nutrient utilization in the soil, enhances soil health, and plays a significant role in plant growth through the secretion of substances like auxin (IAA). In parallel, we observed that applying this strain improved the production efficiency of wheat, corn, and peanuts within complex farming systems, indicating that this method holds the potential for enhancing both the yield and quality of various crops.

**KEYWORDS**  PGPR, growth promote, tobacco, wheat-maize rotation, peanut-maize intercropping

**Peer Reviewer** Sanghamitra Saha, University of Houston-Downtown, Houston, Texas, USA

Address correspondence to Shixiang Zhang, xuyizhangshix@163.com, Weichang Gao, gzyksg@outlook.com, or Ying Jiang, JY27486@163.com.

Shixiang Zhang and Fengmin Li contributed equally to this article. Author order was determined in order of increasing seniority.

The authors declare no conflict of interest.

See the funding table on p. 17.

Plant growth-promoting rhizobacteria (PGPR) play a crucial role in enhancing plant growth and health by directly or indirectly stimulating plant growth. Mechanisms such as nitrogen fixation, phosphorus and potassium solubilization, and production of plant hormones are among the mechanisms through which PGPR promotes plant growth (1). IAA, a vital plant hormone, regulates processes like cell division, elongation, fruit development, and phototropic response (2). Studies indicate that IAA induces root production and increases lateral root formation (3). Phosphorus and potassium are crucial nutrients for plant growth, regulating protein synthesis and stress response processes (4, 5). Despite their natural abundance, phosphorus and potassium bioavailability to plants is often limited in soils (6). Previous studies have demonstrated that PGPR has been shown to modify root structure, enhance nutrient uptake, and improve fertilizer use efficiency, thereby increasing crop yields (7). Fu et al. isolated *Streptomyces lincolnensis* strain L4 from the rhizosphere soil of sweet wormwood herb (*Artemisia annua*) and identified its capacity for hydrolyzing both organic and inorganic phosphorus, nitrogen fixation, and production of IAA. Strain L4 was observed to establish a dominant endophytic bacterial community with a specific structure in plant roots, thereby enhancing root development (8). Additionally, Martinez et al. isolated seven IAA-producing PGPR strains from volcanic soils, all identified as belonging to the genus *Serratia*. Pot experiments showed that these PGPRs have the potential to increase both root diameter and plant height in seedlings of *Nothofagus alpina* (9). Chanyarat Paungfoo-Lonhienne et al. applied PGPR of *Paraburkholderia* to Kikuyu grass (*Pennisetum clandestinum*). The results showed that PGPR could promote crop yield and N fertilizer use efficiency (10). Marta Gallart et al. investigated the effects of a PGPR from the genus *Paraburkholderia* on the growth of Reed avocado (*Persea americana*) seedlings fertilized with inorganic N fertilizer (iN). PGPR increased the nitrogen uptake efficiency of seedlings treated with iN fertilizer by 1.2-fold (11). Nevertheless, the effectiveness of PGPR as a biological agent could be compromised by its limited ability to colonize the rhizosphere (12). Therefore, achieving the same growth-promoting effects observed under laboratory conditions in fields remains a significant challenge.

Tobacco (*Nicotiana tabacum*) is an extremely important economic crop in most countries, particularly in China. The Huang-Huai-Hai Plain is one of the main planting regions. This region is renowned for producing strong aroma flue-cured tobacco varieties. However, due to limited arable land and inadequate scientific farming practices, tobacco farmers frequently encounter challenges associated with continuous cropping. These challenges exacerbate the spread of tobacco autotoxins, impairing root growth and diminishing the quality and yield of tobacco (13, 14). Currently, tobacco production heavily relies on pesticides and fertilizers, which not only escalate production costs but also diminish the quality of tobacco leaves. Gangcai Liu et al. discovered that as the rate of fertilizer application increased, there was a corresponding increase in tobacco yield; however, the net income experienced a substantial decline at an accelerated pace. The opposite was true for fertilizer nutrient use efficiency (15). Thus, it is advisable to adopt an environment-friendly approach to address this issue, which entails prioritizing the utilization of PGPR to enhance fertilizer efficiency (16). Utilizing PGPR as microbial fertilizers provides a solution that balances fertilizer efficiency while ensuring environmental friendliness (17). Siahaan, Parluhutan et al. applied PGPR in combination with NPK fertilizer to tomato (*Solanum lycopersicum* L.) plants and observed that the tomato plants treated with this combination exhibited significant improvements in growth compared with the other treatments (18). Medhat Rehan et al. conducted a screening of multifunctional PGPR in Qassim, Saudi Arabia. They identified four strains, namely *Streptomyces cinereoruber* strain P6-4, *Priestia megaterium* strain P12, *Rossellorea aquimaris* strain P22-2, and *Pseudomonas plecoglossicida* strain P24. The effectiveness of these PGPR strains in promoting plant growth was evaluated through field experiments, in which N, P, and K conventional fertilizers were also applied. The results demonstrated that all bacterial treatments significantly enhanced plant growth and improved phosphorus absorption traits (19). Wang et al. conducted an experiment where they

inoculated wheat (*Triticum aestivum*) with three PGPR strains: *Pseudomonas moraviensis*, *Bacillus safensis*, and *Falsibacillus pallidus*. The objective was to evaluate the impact of these PGPR strains on wheat growth. The results showed that the inoculation of PGPR enhanced the soluble phosphorus content in the soil, promoted wheat growth, and improved fertilizer utilization efficiency. Notably, these positive effects were observed without reducing the amount of fertilizer applied (20). The application of PGPR in tobacco fields is expected to improve fertilizer utilization, crop yield, and quality, but further screening and field trials of tobacco inter-root probiotics are needed to fully utilize their potential.

The Huang-Huai-Hai Plain, known for its diverse agricultural systems, is of great agricultural significance in China. It is a region where tobacco cultivation thrives alongside the prevalent wheat-maize rotation and peanut-maize intercropping, often involving crops that precede or follow tobacco (21). Rotational cropping and intercropping serve as key strategies for enhancing agricultural productivity. These techniques enhance resource utilization, as well as improve the physical, chemical, and biological properties of the soil (22, 23). However, managing the simultaneous cultivation of two different crops in a single field poses greater complexity than cultivating a single crop. Different crops often exhibit distinct nutrient requirements, posing challenges in meeting the optimal nutrient needs of both crops through conventional fertilization methods. Inappropriate fertilizer application can have detrimental effects on agricultural outcomes and the surrounding environment. In addition to reducing fertilizer utilization, these methods result in decreased yields (15). Rhizosphere microorganisms play a crucial role in maintaining soil fertility and ecosystem balance (24). Guo et al. found that the increased relative abundance of *Chlamydiae*, *Saccharibacteria*, and *Parcubacteria* during maize-peanut intercropping enhanced phosphate and hypophosphate metabolism (25), thereby more phosphorus became available for the crops. Rezaei-Chiyaneh et al. confirmed that applying PGPR to fennel (*Foeniculum vulgare*) and common bean (*Phaseolus vulgaris*) intercropping resulted in positive growth-promoting effects (26). Up to date, research on the use of PGPR in rotational and intercropping systems remains limited, possibly due to the complex soil environment associated with these practices (27). It is essential to screen and identify PGPR strains that are well-suited to local farming systems. This not only facilitates the colonization of PGPR strains in local soil environments but also enhances their efficacy in promoting crop growth in practical agricultural production.

The present work aims to uncover the growth-promoting effect of PGPR on tobacco and its adaptability to rotation and intercropping systems. The multifunctional PGPR isolated from the tobacco rhizosphere was examined for its impact on tobacco growth through pot experiments. Furthermore, the broad-spectrum growth-promoting effect of PGPR was investigated through field experiments under wheat-maize rotation and peanut-maize intercropping. This research provides foundational support for implementing PGPR in rotation and intercropping, thus contributing to the development of sustainable agriculture in the Huang-Huai-Hai Plain.

## MATERIALS AND METHODS

### Soil sample and culture substrates

Soil samples were collected from the tobacco planting area in Xuchang City (Henan, China), which were obtained from the alluvial soil in the tobacco rhizosphere at depths of 10–20 cm. All samples were carefully cleaned to remove rocks, stubble, and other debris. All these soil samples were then meticulously sealed in sterilized valve bags and stored at 4°C.

The soil properties were measured as follows: organic carbon content was 6.45 g/kg, total phosphorus content was 0.80 g/kg, available phosphorus content was 9.77 mg/kg, total potassium content was 18.56 g/kg, available potassium content was 111.17 mg/kg, and pH level was 7.27.

The culture media used in this experiment included Luria-Bertani (LB) medium, LB liquid medium, monkina medium, organic phosphorus liquid medium, Pikovskaya (PKO) medium, and the liquid medium for potassium-solubilizing bacteria and inorganic salt medium. For detailed formulations of the culture media, please refer to Table S1.

## Isolation and purification of strains

A single colony of rhizosphere-promoting bacteria was isolated from the tobacco rhizosphere using the dilution method on the plates (28). Soil samples underwent a 1:10 dilution by adding 10 g of soil to a 250 mL conical flask containing 90 mL of sterile water. The flask was then agitated at 30°C and 150 rpm for 20 min. After allowing it to settle, the supernatants were further diluted into 10-fold dilutions ($10^{-2}$, $10^{-3}$, $10^{-4}$, $10^{-5}$, and $10^{-6}$) using sterile water. An appropriate dilution was selected, and then, the bacterial suspension was spread onto an LB solid culture medium. The plates were then incubated at 30°C in a constant temperature incubator for 24 h. After incubation, different types of representative single colonies were selected and subjected to plate purification. Each isolated colony was numbered and stored at 4°C.

## IAA-producing ability of isolated strains

The estimation of IAA through spectrophotometry followed the method developed by Brick et al. (29), with modifications made by Goswami et al. (30). The 24 h culture supernatant of bacterial isolates, obtained from growth in LB liquid medium supplemented with L-tryptophan (100 mg/L) at 30°C and 180 rpm, was mixed in a 1:1 ratio with Salkowski reagent (50 mL, consisting of 35% perchloric acid, and 1 mL 0.5 M $FeCl_3$ solution). The development of a pink color indicated IAA production, and its optical density was measured at 530 nm. The quantity of IAA in the culture filtrate was determined using a standard curve prepared with a known concentration of IAA.

## Organic/inorganic phosphorus solubilization capacity of isolated strains

After isolating and purifying the bacterial strain, a 1% (vol/vol) inoculum was added to 50 mL of either organic phosphorus bacterial liquid culture medium or PKO culture medium. The culture was incubated at 30°C and 180 rpm for 4 days. After incubation, 10 mL of the culture medium was extracted and centrifuged at 10,000 rpm for 15 min. The supernatant was then subjected to the measurement of available phosphorus using the molybdenum blue colorimetric method described by Lu (31), and a standard curve was plotted.

## Morphological, physiological, biochemical, and molecular characterization of selected strains

The selected strains were inoculated on LB plates at 30°C for 24 h, and their colony appearances, including size, shape, color, gloss, consistency, and transparency, were observed under a microscope (SK200, Motic). The morphology and size of the bacterial samples were analyzed using a scanning electron microscope (SEM, S-3400 N, Hitachi) at the Central Laboratory of Henan Agricultural University. Physiological and biochemical characteristics, such as Gram staining, catalase reaction, methyl red test, Voges-Proskauer test, starch hydrolysis test, gelatin liquefaction, nitrate reduction, citrate utilization test, and aerobic test, were determined following standard procedures (32).

## Molecular characterization of selected isolates

The molecular identification of selected strains was done by analyzing 16 S rRNA gene sequences. The universal forward primer 27F and reverse primer 1492R were used to amplify 16 S rRNA genes via PCR (33). Gene sequencing of 16SrRNA was performed at Mei Yi Mei Biological Technology Co., Ltd., Beijing. The 16S rRNA gene sequences were matched with BLAST tool at NCBI (http://www.ncbi.nlm.nih.gov/blast/Blast.cgi)

for bacterial identification and subsequently submitted to NCBI GenBank to obtain accession numbers. The phylogenetic tree was constructed using mega version 7.0 via the neighbor-joining method (34, 35)

## Effect of different culture conditions on the growth and IAA-producing capacity of the isolated tobacco PGPR

To determine the optimal conditions for IAA production by tobacco PGPR, adjustments were made to the liquid culture medium that included L-tryptophan (100 mg/L). These adjustments involved varying the liquid volume (25, 50, 75, 100, and 150 mL of LB liquid medium in 250 mL Erlenmeyer flask), pH levels (4, 5, 6, 7, 8, 9, and 10), carbon sources (glucose, xylose, sucrose, fructose, mannitol, lactose, and maltose), and nitrogen sources (ammonium nitrate, ammonium sulfate, potassium nitrate, peptone, urea, yeast extract, and glutamate). The selected strains were inoculated at a 1% (vol/vol) volume into 250 mL Erlenmeyer flasks and incubated at 30°C on a shaking incubator at 180 rpm for 24 h. IAA production was quantitatively measured (30, 36), and the growth of the strains was assessed by measuring the optical density at 600 nm (37).

## Pot and field experiments of the selected PGPR on tobacco growth

Before planting, the selected strain was separately inoculated into 250 mL Erlenmeyer flasks containing 100 mL of LB liquid medium and incubated at 30°C with shaking (180 rpm) for 48 h. The bacterial cultures were centrifuged at 10,000 rpm for 10 min. The collected bacterial cells were then resuspended in sterile water to adjust the concentration to $10^{11}$ CFU/mL for inoculation, producing bacterial inoculants. Soil samples were collected from the topsoil (0–20 cm) from farmland near the site of the field experiment. After the soil was air-dried, the impurities were removed by passing the soil through a 5 mm sieve. Tobacco seedlings were obtained from the Zhengzhou Tobacco Research Institute of the China National Tobacco Corporation. In total, 0.7 kg of prepared soil was placed into each pot. Tobacco seedlings with identical growth conditions were selected for transplanting, with each pot containing one plant. Two different treatments were designed as follows: (i) CK, tobacco plants were inoculated with an equal volume of sterile water (control), and (ii) L8, tobacco plants were inoculated with bacterial inoculants at an inoculation rate of $10^8$ CFU/g dry soil. Each treatment was independently replicated three times, and the potted plants were transferred to the greenhouse. The greenhouse was set up as a light/dark period of 16/8 h, a relative humidity of 60% ± 5% with a temperature of 25°C. Deionized water was added to maintain the moisture concentration at approximately 60% of the moisture-holding capacity.

After 30 days, the plastic pot was carefully cut open to remove the soil and plant together. The plant was gently uprooted and shaken to separate the loose soil. The entire soil mixture was thoroughly mixed, and a representative sample was taken to accurately reflect the overall soil condition. This sample was then sieved through a 2 mm mesh, and a portion of the sieved sample was stored in a refrigerator at 4°C for further analysis. The quantification of IAA was determined by high-performance liquid chromatography (HPLC) (38). The available phosphorus and potassium levels in the soil were determined using the molybdenum antimony anti-colorimetric method and flame photometry, respectively (31).

Tobacco plants were washed thoroughly with tap water to remove the soil from the roots and subsequently preserved in 70% alcohol. Root images were obtained using a scanner (LA1600 + scanner, Canada), and analysis of root-related parameters (total length, average diameter, total surface area, and number of tips) was performed using Win-rhizo software (Win-rhizo2003b, Canada). The root system was identified into five categories based on root diameter (RD): I (RD 0–0.5 mm), II (RD 0.5–1.0 mm), III (RD 1.0–1.5 mm), IV (RD 1.5–2.0 mm), and V (RD >2.0 mm).

The fresh weight, plant height, relative chlorophyll content, and total nitrogen, phosphorus, and potassium content of the plants were quantified. The measurements of the aforementioned indicators were referenced from the Methods of Soil Agricultural

Chemistry Analysis (27). Cured tobacco leaves from the middle part of the plant were collected for the purpose of determining the content of nicotine, total sugar, reducing sugar, potassium, and chlorine. Nicotine content was determined by gas chromatographic analysis, total sugar, and reducing sugar were determined by the anthrone-sulfuric acid method, whereas total nitrogen, potassium, and chlorine content were measured using a flow analyzer (31, 39).

The tobacco field experiment was conducted in the Xuchang Experimental Zone, where the soil type was sandy flavor-aquic soil, and the physicochemical properties of the soil were the same as mentioned above. The tobacco variety was identified as "Yuyan10." The field was cultivated following local tobacco farming practices, including soil preparation, application of base fertilizers, ridging, and transplanting. The local conventional fertilization was applied with N 54.22 kg/hm$^2$, P$_2$O$_5$ 95.85 kg/hm$^2$, and K$_2$O 347.26 kg/hm$^2$, with 70% applied as basal fertilizer and 30% as top dressing. The experiment variables were limited to strain inoculation. On the basis of conventional fertilization, the treatment group received the selected strain formulation, which was combined with a sterilized bone meal carrier, at a viable count of 10$^{11}$ CFU/g and a fertilizer application rate of 40 kg/hm$^2$. The control group was supplemented with inactivated microbial agents, using sterilized bone powder as the carrier, in addition to conventional fertilization. Each treatment plot covered 50 m$^2$, replicated three times in a randomized block design. At the end of the experiment, plants were harvested, and a five-point sampling method was used to randomly select 10 tobacco plants from each plot. The nicotine content, total sugar, and reduced sugar, potassium, and chlorine content in the plant samples were analyzed. The yield of cured tobacco leaves and the proportion of different grades were calculated, and the final yield was determined accordingly. The 2 kg soil samples were collected from the 0–20 cm depth around the roots of the tobacco plants, thoroughly mixed, and pretreated for soil indicator measurements. Soil indicators including pH, organic matter, available nitrogen, available phosphorus, and available potassium were measured (31, 39).

## Field experiments of the selected PGPR on plant growth in wheat (*Triticum aestivum*)-maize (*Zea mays*) rotation and peanut (*Arachis hypogaea*)-maize intercropping

The experiment was conducted at the North China Wheat-Maize Rotation Nutrition and Fertilization Scientific Field Experimental Station in Xinzheng City (Henan, China). The soil type was sandy loam, and the basic soil properties were as follows: organic carbon content of 1.91 g/kg, total phosphorus content of 0.29 g/kg, available phosphorus content of 3.44 mg/kg, total potassium content of 19.56 g/kg, available potassium content of 20.42 mg/kg, and pH value of 7.39.

Winter wheat was sown in early October and harvested in early June. The tested wheat variety was "Aikang 58." The sowing rate was 150 kg/hm$^2$. The local conventional fertilization method was used, with a basal application of compound fertilizer (N: P$_2$O$_5$: K$_2$O = 15:15:15) at a rate of 600 kg/hm$^2$. During the tillering stage of wheat, urea was applied through furrow irrigation at a rate of 120 kg/hm$^2$. The experimental setup for both the treatment group and the control group followed the same as the above tobacco field experiment. Other management measures during the growth period were the same as those for high-yield wheat fields in the local area. After wheat harvest, the spike number per meter of the double row was recorded. Five-point sampling method was used to set up sampling points in each plot. Two wheat plants were randomly selected from each point, resulting in a total of 10 plants per plot, to measure aboveground biomass, effective panicles number, grain number per spike, and 1,000-grain weight. Actual yield was measured in a 4 m$^2$ area within each plot.

Maize and peanuts were planted concurrently in early June and harvested in mid-September. The corn variety tested was "Yudan 9953," and the peanut variety was "Yuhua 9719." Maize and peanuts were intercropped in a 2:4:2 pattern, with 40 cm between maize rows, 30 cm between peanut rows, and 60 cm between maize and

peanut rows. The experiment comprised 10 plots, each covering an area of 50 m$^2$. Compound fertilizer (N: P$_2$O$_5$: K$_2$O = 15:15:15) was applied as basal fertilizer at a rate of 650 kg/hm$^2$, without additional fertilization. The treatment and control groups were set up following the methods used in the tobacco field experiment mentioned above, and the field was irrigated using the local cultivation method. After corn and peanut harvest, two rows each 2 m in length were selected to measure biomass production. In each plot, 10 corn plants were selected to measure ear length, grain number per row, 100-grain weight, single cob weight, and ear bald length; 10 peanut plants were sampled to measure the number of filled fruits per plant, the number of unfilled fruits per plant, the weight of filled fruits per plant, and the weight of unfilled fruits per plant.

## Statistical analysis

All statistical analyses were conducted using SPSS 16.0 (SPSS Inc., Chicago, IL, USA). The significance of differences among treatments was evaluated using a one-way ANOVA with a least significant difference (LSD) test ($P < 0.05$). Pearson's correlation analysis was used to investigate the relationships between various indicators. Before conducting principal component analysis (PCA) using Metabo Analyst 5.0, all data were log-transformed. ClustVis was used to generate PCA plots and heat maps. Origin 2018 (Origin Lab Corporation, Northampton, MA, USA) was used to generate all graphs.

## RESULTS

### Isolation of strains from tobacco rhizosphere

Fifteen strains of PGPR were isolated and purified from the tobacco rhizosphere soil at the National Tobacco Base of Xuchang City. To identify potential growth-promoting characteristics among the isolated bacteria, we assessed a range of pertinent indicators. The screening results for indole-3-acetic acid (IAA) production showed that six isolate strains, namely L2, L3, L6, L8, L10, and L13, exhibited the capability to produce IAA. In the K solubilization test, eight strains (L2, L4, L5, L8, L10, L12, L13, and L15) demonstrated the ability to solubilizing K. In the organic P solubilization test, eight strains (L1, L3, L7, L8, L9, L11, L13, and L15) exhibited the capacity to solubilize organic P. In the inorganic P solubilization test, six strains (L1, L6, L8, L9, L13, and L15) showed the ability to solubilize inorganic P (Fig. 1). Various strains exhibited differing degrees of growth-promoting characteristics. Among these strains, L8 exhibited the highest IAA production, reaching 61.71 µg/mL. Its K solubilization, organo-P, and P solubilization capacity were 19.88 mg/L, 1. 57 mg/L, and 221. 65 mg/L. Strain L8 was chosen for further investigation owing to its high IAA production capacity and significant potassium and phosphate solubilization abilities.

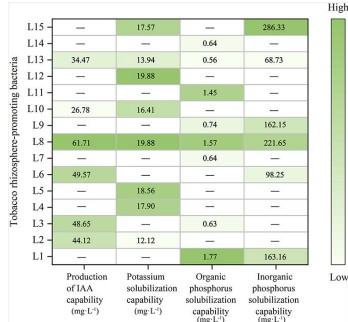

FIG 1  The abilities of 15 different tobacco rhizosphere-promoting bacteria in terms of IAA production, potassium solubilization, organic phosphorus solubilization, and inorganic phosphorus solubilization. The varying intensity of colors denotes the strength of representation, and the deeper the color, the greater the potency of the ability.

## The morphological, physiological, biochemical, and molecular characterization of strain L8

To identify strain L8, morphological and biochemical tests were conducted. Light microscopy of selected bacterial isolates (strain L8) revealed rod-shaped and Gram-positive cells. Fig. 2A showed that most bacterial colonies were light yellow, round, and opaque, with irregular margins and convex surfaces. The isolates underwent characterization via various biochemical reactions such as contact enzyme test, methyl red (MR) test, Voges-Proskauer (VP) test, starch hydrolysis, gelatin liquefaction, nitrate reduction, and citrate utilization to assess their phenotypic properties. The results indicated that L8 was a facultative anaerobe. It showed positive results in Gram staining, catalase reaction, starch hydrolysis test, gelatin liquefaction, and nitrate reduction, whereas testing negative in methyl red test, Voges-Proskauer test, and citrate utilization test (Fig. 2B). 16S rRNA gene sequencing of strains L8 revealed high sequence similarity with *Bacillus thuringiensis* (NR 112780). Strain L8 was submitted to the GenBank nucleotide sequence database and allocated with accession number OR545797. Figure 2C shows the 16S rRNA sequence-based phylogenetic tree of strain L8.

## The growth and IAA production ability of the L8 strain under different culture conditions L8

The growth was found to be optimum when the liquid volume of LB liquid medium in 250 mL Erlenmeyer flask was 25 mL. The effect of pH on the growth of L8 was measured at a range of 4.0–10.0, and the maximum growth at pH 8.0. The optimal carbon sources for the growth of L8 were found to be mannitol, followed by glucose and sucrose, whereas yeast extract was identified as the best nitrogen source. Isolate L8 exhibited the highest production of IAA, with concentrations of 75 mg/L at 50 mL and 66 mg/L at 75 mL liquid volume, respectively. At pH 6, L8 produced 70.87 mg/L of IAA, which decreased to 70.58 mg/L at pH 7, with further decreases observed with higher pH levels. The isolate exhibited varying IAA production rates across different carbon sources, with the highest at fructose, followed by sucrose, mannitol, and glucose, whereas others remained below 45 mg/L. Glutamate was identified as the optimal nitrogen source for strain L8, showing an IAA production of 75.43 mg/L, followed by peptone (Fig. S1).

## Effect of L8 inoculation on soil nutrient indicators, plant growth, and root architecture of tobacco in pot experiment

In pot experiments, tobacco inoculated with L8 exhibited significant growth-promoting effects. Treatment with L8 increased IAA levels by up to 1.64-fold in the soil, whereas the concentrations of available phosphorus and potassium increased by 33.93% each. Compared with control group, root length, surface area, and volume increased significantly across all classes by up to 3.4-fold with L8 inoculation (Table 1). Inoculation of tobacco plants with L8 also yielded notable outcomes. Results indicated significant increases in plant fresh weight, soil plant analysis development (SPAD), and height due to L8 treatment. Analysis of variance showed higher plant N, P, and K levels in L8-inoculated plants compared with controls, with increases of 15.05%, 23.78%, and 30.81%, respectively (Table 1).

## Principal component analysis, heatmap analysis, and correlation matrix of soil nutrients, root architecture, and plant growth in tobacco pot experiment

PCA was conducted to assess the influence of inoculating strain L8 on multiple indices of soil nutrients, root system configuration, and plant growth in potted tobacco experiments. PCA score plots indicated that PC1 and PC2 contributed 90.9% and 4.2%, respectively, to the total variance (Fig. 3A). For tobacco soil nutrients index, the highest value of mean decrease accuracy was IAA; for root system configuration index, the highest value of mean decrease accuracy was RV I; for plant growth index, the highest value of mean decrease accuracy was plant K (Fig. 3B). Correlation analysis revealed

**TABLE 1** The effects of L8 bacterial strain inoculation (CK: conventional treatment; L8: inoculation of L8 strain based on conventional treatment) on tobacco soil IAA levels and nutrient indicators (soil available *P*; soil available K), root system architecture, and classification, physiological and nutrient indicators (plant fresh weight; plant height; SPAD; plant N; plant *P*; plant K) of plant growth in the potted plant experiment[a]

| Indicators | Projects | CK | L8 |
|---|---|---|---|
| Soil IAA levels and nutrient indicators | Soil IAA (mg·kg$^{-1}$) | 0.25 ± 0.04 | 0.66 ± 0.07** |
| | Soil available P (mg·kg$^{-1}$) | 5.04 ± 0.08 | 6.75 ± 0.42* |
| | Soil available K (mg·kg$^{-1}$) | 132.00 ± 4.24 | 149.00 ± 1.73** |
| Root system architecture and classification | Root length (cm) | 297.00 ± 59.45 | 757.09 ± 146.49** |
| | I (RD 0–0.5 mm) | 245.50 ± 54.15 | 609.34 ± 143.23* |
| | II (RD 0.5–1.0 mm) | 40.73 ± 5.26 | 117.79 ± 9.90** |
| | III (RD 1.0–1.5 mm) | 5.64 ± 1.48 | 14.49 ± 0.52** |
| | IV (RD 1.5–2.0mm) | 1.71 ± 0.25 | 5.85 ± 1.96* |
| | V (RD >2.0 mm) | 3.35 ± 0.54 | 9.48 ± 1.14** |
| | Root surface area (cm$^2$) | 39.19 ± 6.66 | 108.97 ± 12.53** |
| | I (RD 0–0.5 mm) | 19.04 ± 4.51 | 50.65 ± 12.81* |
| | II (RD 0.5–1.0 mm) | 8.15 ± 0.92 | 23.85 ± 2.00** |
| | III (RD 1.0–1.5 mm) | 2.14 ± 0.57 | 5.43 ± 0.12** |
| | IV (RD 1.5–2.0mm) | 0.93 ± 0.11 | 3.12 ± 1.07* |
| | V (RD >2.0 mm) | 3.64 ± 0.79 | 12.11 ± 1.32** |
| | Root volume (cm$^3$) | 0.41 ± 0.06 | 1.25 ± 0.06** |
| | I (RD 0–0.5 mm) | 0.14 ± 0.04 | 0.38 ± 0.10* |
| | II (RD 0.5–1.0 mm) | 0.14 ± 0.01 | 0.40 ± 0.03** |
| | III (RD 1.0–1.5 mm) | 0.07 ± 0.02 | 0.16 ± 0.00** |
| | IV (RD 1.5–2.0mm) | 0.04 ± 0.00 | 0.13 ± 0.05* |
| | V (RD >2.0 mm) | 0.03 ± 0.01 | 0.17 ± 0.05* |
| | Average root diameter (mm) | 0.42 ± 0.02 | 0.46 ± 0.03 |
| | Number of root tips | 687.00 ± 206.28 | 995.67 ± 180.89 |
| | Number of root forks | 1478.00 ± 281.86 | 3608.33 ± 422.57** |
| Physiological and nutrient indicators of plant growth | Plant fresh weight (g) | 5.47 ± 1.80 | 10.02 ± 2.11** |
| | Plant height (cm) | 17.12 ± 1.83 | 22.14 ± 1.77** |
| | SPAD | 3.35 ± 0.16 | 3.83 ± 0.14** |
| | Plant N (g·kg$^{-1}$) | 0.93 ± 0.09 | 1.07 ± 0.08* |
| | Plant P (g·kg$^{-1}$) | 1.64 ± 0.19 | 2.03 ± 0.11** |
| | Plant K (g·kg$^{-1}$) | 24.08 ± 1.66 | 31.50 ± 2.20** |

[a]Note: N, nitrogen; P, potassium; K, phosphorus; SPAD, relative chlorophyll content; RD, root diameter. *, indicates significant differences at $P < 0.05$; **, indicates extremely significant differences at $P < 0.01$. The root architecture indexes were divided into five categories based on root diameter: I (RD 0–0.5 mm), II (RD 0.5–1.0 mm), III (RD 1.0–1.5 mm), IV (RD 1.5–2.0 mm), and V (RD >2.0 mm).

significant positive correlations: soil available P with RL II, RL IV, RL V, RV II, RV IV, RSA II, RSA IV, and Plant P; soil available K with Plant P, Plant K, Plant Height, and SPAD; soil IAA with RL II, RV I, RV II, RSA, and RSA II ($P < 0.05$) (Fig. 3C and D).

## Effects of L8 inoculation on the physicochemical properties of tobacco field soil, tobacco yield, and quality in the field experiment

In the field experiment, inoculation with L8 significantly increased tobacco soil nutrient content, effectively promoted the yield, and improved the quality of flue-cured tobacco (Fig. 4). Compared with the CK treatment, L8 treatment significantly increased the organic matter, available phosphorus, and available potassium by 65.89%, 12.84%, and 14.07%, respectively ($P < 0.05$), the available nitrogen content increased by 2.79%. In middle leaves, L8 treatment increased total sugar, reducing sugar, nicotine, K$^+$, K$^+$/ Cl$^-$ ratio, sugar/nicotine ratio, and total sugar/reducing sugar ratio by 7.69%, 17.33%, 5.06%, 18.24%, 41.95%, 11.67%, and 9.64% relative to CK, L8 also decreased Cl$^-$ content in middle leaves, to a great extent. The yield of flue-cured tobacco and the proportion of high-quality tobacco were increased by 6.60% and 5.90%, respectively.

(A)

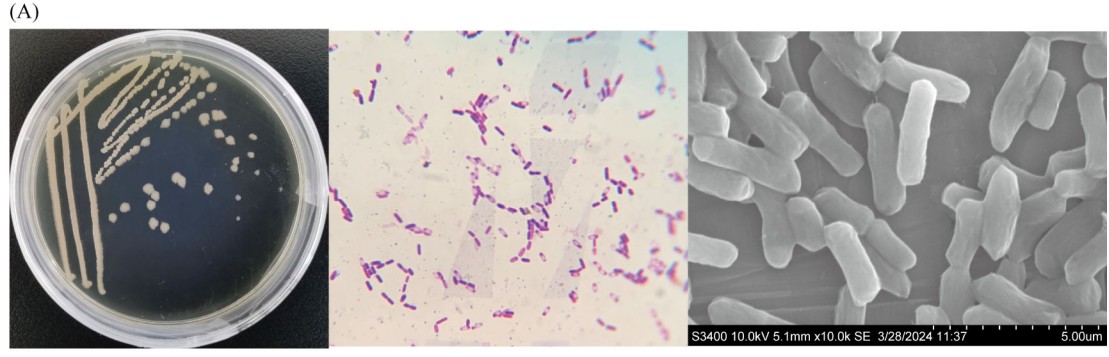

(B)

| Projects | Results | Projects | Results |
|---|---|---|---|
| Gram staining | + | Starch hydrolysis test | + |
| Aerobic test | Facultative anaerobic | Gelatin liquefaction | + |
| Catalase reaction | + | Nitrate reduction | + |
| Methyl red test | - | Citrate utilization test | - |
| Voges-Proskauer test | - | | |

Note: "+" indicates positive reaction; "-" indicates negative reaction.

(C)

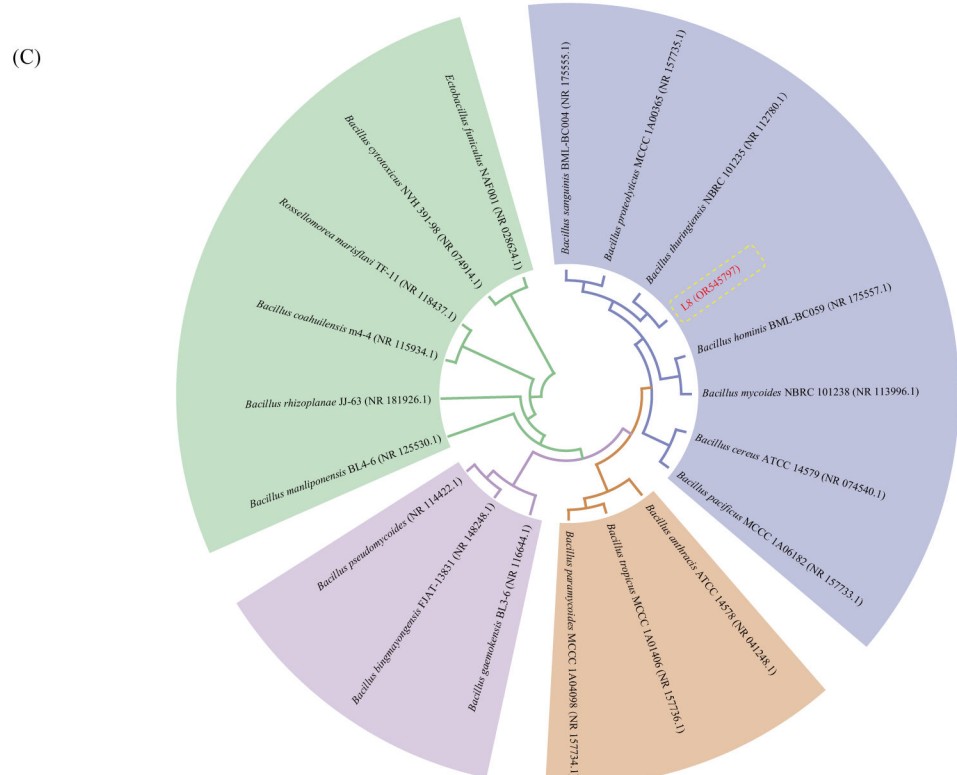

FIG 2  The morphological, physiological, biochemical, and molecular characterization of strain. The morphology of the bacterial colonies, Gram stain reaction, and imaging by SEM ($\times10^4$) of L8 strain (A); Physiological and biochemical characteristics of L8 strain (B); phylogenetic tree of L8 (OR545797) based on the 16S rRNA gene sequence, two substitutions per 1,000 nucleotide positions (C).

## Effects of L8 inoculation on the soil physicochemical properties and yield of wheat, maize, and peanut in the field experiment

Inoculation with L8 not only influenced the content of soil nutrients in wheat, corn, and peanut fields but also enhanced the yield of crops. Under wheat field conditions,

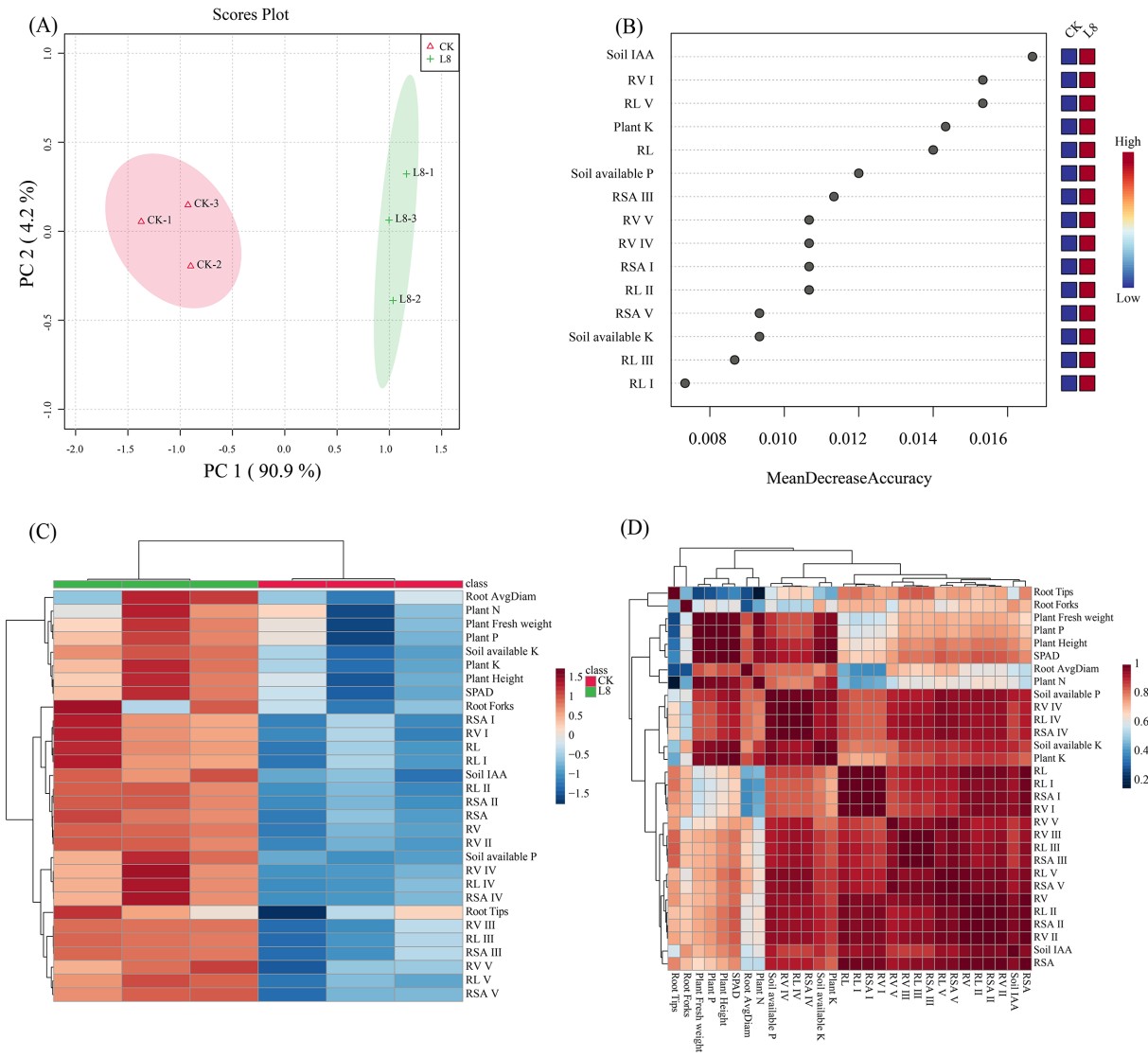

**FIG 3** PCA (A) random forest analysis (B) correlation matrix displaying correlation (C), and heatmap analysis (D) of different treatments (CK: conventional treatment; L8: inoculation of L8 strain based on conventional treatment) on soil nutrients, root architecture, and plant growth in the tobacco pot experiment. PC1: first principal component; PC2: second principal component. RL, root length, RSA, root surface area; RV, root volume. The root system was divided into five categories based on root diameter (RD): I (RD 0–0.5 mm), II (RD 0.5–1.0 mm), III (RD 1.0–1.5 mm), IV (RD 1.5–2.0 mm), and V (RD >2.0 mm).

compared with the CK treatment, L8 treatment significantly increased soil organic matter, organic N, available P, and available K contents by 39.11%, 5.38%, 13.10%, and 16.92%, respectively ($P < 0.05$) (Fig. 5A). Additionally, L8 treatment significantly increased the aboveground biomass, effective panicle number, thousand-grain weight at the maturity, and wheat yield by 76.04%, 76.28%, 7.59%, and 40.51%, respectively ($P < 0.05$) (Fig. 5B). Similarly, under maize field conditions, L8 treatment significantly increased soil organic matter, available P, and available K contents by 77.48%, 20.76%, and 15.26%, respectively ($P < 0.05$) (Fig. 5C). Meanwhile, L8 treatment significantly increased the ear length, grain number per row, and Single cob weight of maize by 3.93%, 17.11%, and 11.25%, respectively, while decreasing ear bald length by 27.57%. Maize yield significantly increased by 13.31% ($P < 0.05$) (Fig. 5D). In peanut fields, L8 treatment significantly increased the soil organic matter, organic N, available P, and available K contents by 58.56%, 27.75%, 32.32%, and 45.51%, respectively ($P < 0.05$) (Fig. 5E). Furthermore, application of L8 effectively reduced the incidence of unfilled fruits in peanuts; compared

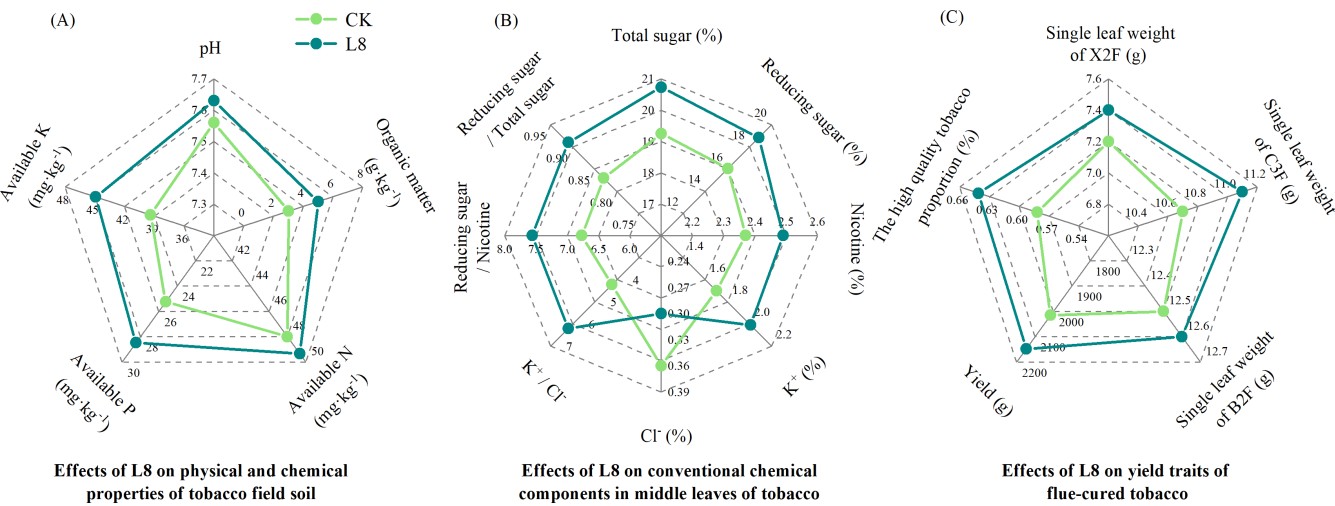

FIG 4 The effects of different treatments (CK: conventional fertilization; L8: application of L8 strain with the bone meal as a carrier on top of conventional fertilization) on tobacco field soil physicochemical properties (A) the conventional chemical composition of middle leaves of tobacco plants (B) and tobacco yield traits (C) in field experiments.

with CK treatment, it significantly decreased the number and weight of unfilled fruits per plant by 47.87% and 42.56%, respectively, and the yield of peanut was significantly increased by 15.19% ($P < 0.05$) (Fig. 5F).

## DISCUSSION

In this study, 15 PGPR strains designated as L1-15 were isolated from tobacco rhizosphere soil. All isolates showed at least one of the assessed activities, including IAA synthesis, potassium solubilization, and solubilization of organic and inorganic phosphorus; among these, isolate L8 demonstrated the highest ability to synthesize IAA. Isolate L8 exhibited specific capabilities: IAA production (61.71 mg/L), potassium solubilization (19.88 mg/L), organic phosphorus solubilization (1.57 mg/L), and inorganic phosphorus solubilization (221.65 mg/L). The promotion in the growth and development of plants by these PGPR isolates may be due to different mechanisms of action like production of plant hormones and increased nutrient uptake. The halotolerant bacterium *Enterobacter* sp. strain P23, isolated from rice fields in India, has found high ACC deaminase activity, phosphorus solubilization, and production of IAA, Sid, and HCN (40). Under non-stress conditions, rice seedlings inoculated with P23 exhibited improved morphological parameters: shoot length, root length, shoot fresh weight, and root fresh weight. Awad et al. (41) reported that bacterization of maize (*Zea mays* L.) plants with *Azotobacter chroococcum*, a strain of EPS-producing bacteria containing ACC-deaminase, increased concentrations of N, P, and K in plant tissues and alleviated salinity stress (41). Rana et al. evaluated the effects of multifunctional PGPR identified as *Brevundimonas diminuta* (AW7) and observed increased biomass in PGPR-inoculated rice (42). It has been frequently reported that an effective biological control or biofertilizer strain isolated from one region may not perform similarly in other soil conditions (43). Therefore, the selection of an effective PGPR was based on a wide range of attributes, making them adaptable to diverse environments, tillage systems, and soil types. We evaluated the growth-promoting effect of L8 in pot experiments and field experiments under different tillage systems.

The application of PGPR in agricultural fields is severely limited due to a wide range of ecological factors that impact bacterial survival and activity (44). Despite numerous reports on PGPR, they have not been still commercialized efficiently. An optimal PGPR should exhibit high rhizosphere competence, environmental friendliness, significant colonization of plant roots upon inoculation, and plant growth-promoting capabilities

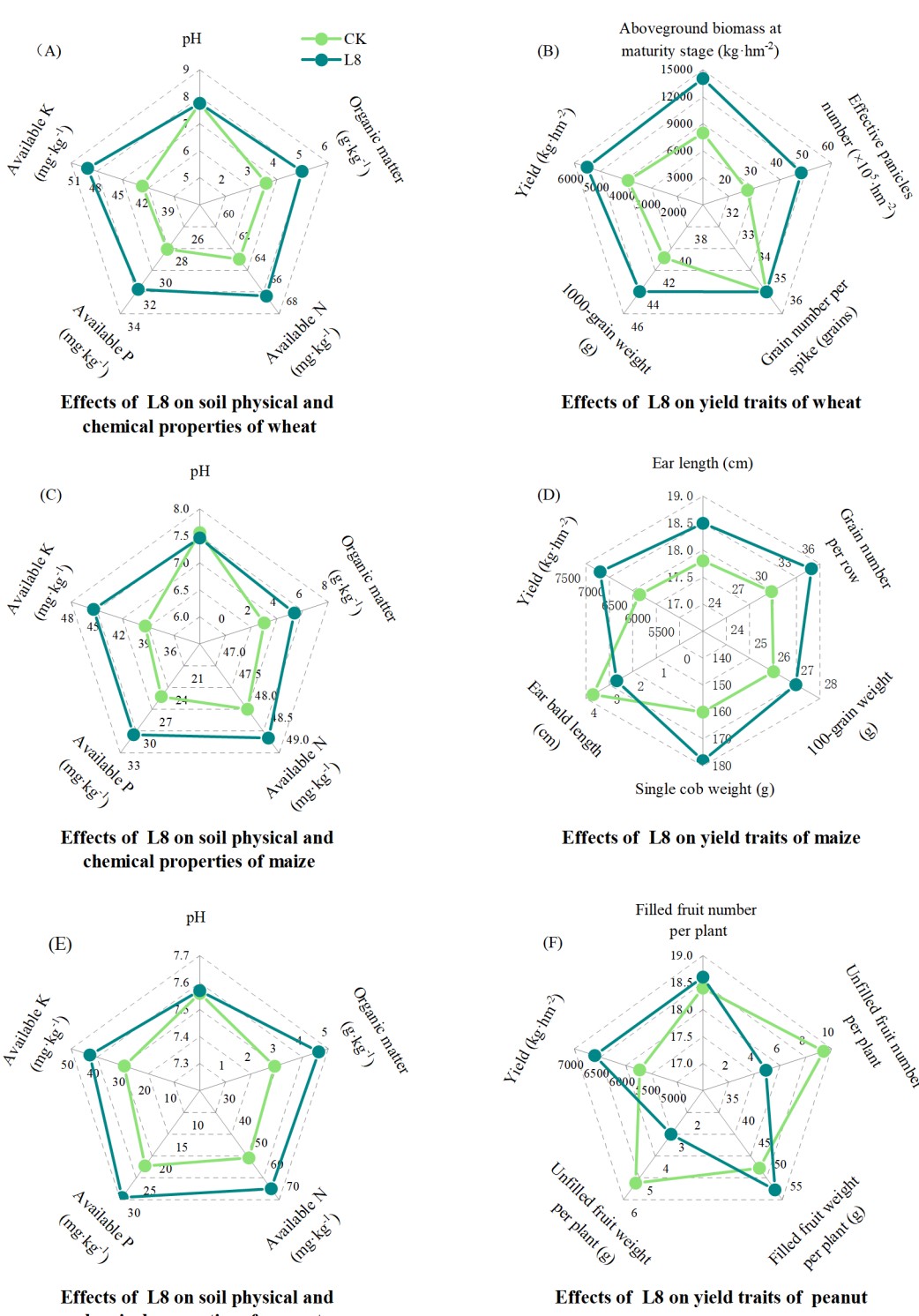

**FIG 5** The effects of different treatments (CK: conventional fertilization; L8: application of L8 strain with bone meal as carrier on top of conventional fertilization) on soil physicochemical properties of wheat (A) maize (C), and peanut field experiment (E) yield and yield components of wheat (B) maize, (D) and peanut field experiment (F).

(45). In this study, the L8 strain grew well when the soil pH was within the range of 6–9. The strain could use a variety of carbon and nitrogen sources, with optimal growth observed when mannitol is employed as the carbon source and yeast extract as the nitrogen source. The L8 strain's ability to survive in adequate amounts across various

carbon and nitrogen sources and a wide pH range is a crucial characteristic for its practical application in agricultural production. The IAA synthesis is one of the important criteria for PGPR screening. It is involved in different processes and controls various physiological functions of plants, including stimulation of seed germination, plant cell division and cell elongation, apical dominance, phototropism, and growth rate, and also controls the lateral and adventitious root formation (2, 46). IAA production is among the most important mechanisms used by PGPR to enhance plant growth, health, and productivity. The production of this hormone is influenced by culture conditions and the availability of substrates in the culture medium (47). L8 was able to produce a certain amount of IAA across the tested pH range. The maximum IAA production in the L8 strain was observed in the pH range of 6–7, which then decreased with an increase in pH. For isolate L8, the best carbon source for IAA production was found to be fructose, followed by sucrose, mannitol, and glucose. The best nitrogen source for IAA production was found to be glutamate and peptone. There is a significant correlation between bacterial growth and IAA production. Acidic or highly alkaline pH levels are not suitable for IAA production due to poor bacterial growth (48). Similar to our findings, Srisuk et al. demonstrated optimal IAA production at pH 6.0 in *Enterobacter* sp. DMKU-RP206 (49). Madhuri also showed maximum IAA production at pH 7.0 in *Rhizobium* strains (50). The pH variations may severely affect both plants and microorganisms, influencing numerous physiological and metabolic processes occurring within the rhizosphere (47). Notably, the pH conditions conducive to L8 growth and high IAA yield are comparable with the pH conditions of the soil in the Huang-Huai-Hai Plain. The L8 strain possesses favorable traits for practical agricultural applications.

The results of the pot experiment show that after inoculating with L8, the soil IAA content, available phosphorus content, and available potassium content increased significantly by up to 1.64-fold, 33.39%, and 12.88%, respectively. Potassium can enhance the flammability of tobacco leaves (51–53). The overall survival of plants depends on the development, growth, and functioning of their roots. Root development and growth are influenced not only by genetic programming but also by phytohormones, nutrients, and environmental factors, to which roots adapt (54). PGPR can alter root-system architecture by producing phytohormones, notably IAA. IAA induces the differentiation of dry tissue into root tissue by influencing plant cell division and promotes longer roots with increased numbers of root hairs and lateral roots (55). Root morphology shows varied response mechanisms at different levels of phosphorus and potassium. It was the fine root growth that was first inhibited under severe P deficiency (56). Appropriate K supply increases root surface area, which plays a critical role in plant growth (57). After inoculation with L8, the root morphology of tobacco plants improved a lot. Most of the root parameter values of inoculated plants were significantly higher than those of uninoculated controls. It is worth noting that the significant increase in the number of root bifurcations provides strong evidence that inoculation with L8 resulted in a substantial increase in lateral roots. Extended lateral roots facilitate enhanced water and nutrient absorption after rainfall, or to survive in shallow soil areas, allowing plants to find the highest nutrient concentrations in shallow soil (58). A number of previous studies have found that inoculation with PGPR promotes lateral root growth. For instance, the inoculation of *Arabidopsis thaliana* plants with *Bacillus megaterium* induced lateral root growth development, increased lateral root number, and promoted root hair length (59). Recently, the research group of Chu et al. also found that *Pseudomonas* PS01 triggered the formation of lateral roots and altered the root system architecture (RSA) (60). A greater number of branches may be positively correlated with higher photosynthetic rates and transpiration efficiency because of the availability of adequate nutrients, compared with non-inoculated plants (61). Therefore, the inoculation with L8 improved the root morphology of tobacco and enabled it to better absorb nutrients. The results of pot experiments also indicate that following inoculation with L8, tobacco plants showed a highly significant increase in fresh weight, plant height, relative chlorophyll content, total phosphorus content, and total potassium content. The strain L8 enhances the

release of available nutrients in the soil, activating some fixed phosphorus and potassium compounds to make them more easily absorbable by tobacco, thereby promoting the growth of tobacco roots and aboveground parts. PGPR has been reported to increase P availability by organic acids that convert insoluble P to plant-available soluble form (62). However, there are few studies on screening multifunctional PGPR from tobacco rhizosphere. It is believed that a variety of small mechanisms of PGPR operate simultaneously or continuously, and the influence of these small mechanisms will have a greater ultimate impact on plants. We believe that PGPR with multiple functions may exert more and better growth-promoting effects than PGPR with only one growth-promoting function (63). Our research offers evidence that alongside nutrient release, L8 enhanced the morphology of the tobacco root system, leading to well-developed tobacco roots that are more effective in nutrient absorption, consequently boosting biomass and tobacco growth.

The potential of biofertilizers still remains untapped despite the benefits. The major challenge in utilizing these formulations to enhance crop productivity is the variability and inconsistency of experimental results across laboratory, greenhouse, and field settings (45). Zhang et al. studied bacteria from tobacco rhizosphere soil and found that strains XF11 and JM3 significantly enhanced seedling height, dry weight, and the absorption of N and P. However, these results were obtained in a greenhouse environment, and their applicability to field conditions may vary (64). To verify the growth-promoting effect of L8 in field conditions, our field experiments demonstrated that inoculation with L8 significantly enhanced soil nutrient content, thereby increasing yield and improving the quality of flue-cured tobacco. Indira Devi et al. highlighted the distinctness of indigenous fluorescent *Pseudomonas* isolates (FPIs) compared with reference fluorescent *Pseudomonas* strains (FPSs). The endemic rhizobacterial pool in the broad bean fields of Imphal Valley contains FPIs that are superior to exotic FPSs (65). Indigenous bacteria may exhibit superior growth-promoting function and stress resistance compared with exotic strains. In this study, L8 was selected from the Huang-Huai-Hai Plain and applied in the local soil environment, facilitating its long-term and stable colonization in the local crop rhizosphere. A more focused study to determine the appropriate PGPR strains, and field testing of potential candidates is important. This will enhance the progress of the selection of better strains as bioinoculants in the field (45).

Some growth-promoting strains can not only survive in the rhizospheric environment of their original host plants but also colonize and demonstrate excellent growth-promoting effects in the rhizospheric soil of other plant species. Shang et al. screened *Bacillus cereus*, *Bacillus methylotrophicus*, and *Bacillus amyloliquefaciens* from the tomato (*Solanum lycopersicum*) rhizosphere and inoculated them into tobacco seedlings under salt stress. The results indicated that upon inoculation with these strains, the effectiveness of soil to plants was significantly improved and mitigated salt-induced damage to tobacco plants (66). He et al. applied a strain of *Bacillus subtilis* isolated from the rhizosphere of *Haloxylon ammodendron* to perennial ryegrass (*Lolium perenne*) and found that this strain enhanced the drought tolerance of perennial ryegrass, promoting its growth and root development (67). The cropping systems of the Huang-Huai-Hai Plain are diverse, consisting primarily of wheat-maize rotation and peanut-maize intercropping. To assess the broad-spectrum efficacy of strain L8 and its adaptability within the complex cropping systems of this region, L8 was applied in field experiments encompassing both wheat-maize rotation and peanut-maize intercropping. The results indicated that L8 significantly enhanced the organic matter, available phosphorus, and available potassium content in the field soil. It resulted in significant yield increases for wheat, maize, and peanuts, with improvements of 40.51%, 13.31%, and 15.19%, respectively. Therefore, L8 not only demonstrated significant growth-promoting effects on wheat, maize, and peanuts but also exhibited broad-spectrum efficacy. It has excellent application potential in both intercropping and rotation systems. There has been relatively limited research on the application of the broad-spectrum characteristics of PGPR to crop rotation and intercropping systems. This is due to the complexity and

diversity of environmental factors in field trials involving intercropping and crop rotation, which makes it challenging for PGPR to exert its growth-promoting functions. Previous studies have demonstrated that agricultural systems under different cultivation practices exhibit significant differences in autochthonous bacterial populations (68–70). Different crop types and management practices can lead to changes in soil physical structure, chemical properties, and enzyme activity, which in turn result in variations in bacterial communities. The performance of PGPR across different tillage systems is influenced by various environmental factors, which need to be considered comprehensively to evaluate its growth-promoting effect. It remains possible to achieve significant growth-promoting effects in complex field environments by screening and applying PGPR strains with strong adaptability and diverse functions.

In this study, L8 isolates with multiple plant growth-promoting properties were obtained from the tobacco rhizosphere, demonstrating growth-promoting effects across various tillage systems (Fig. 6). Trabelsi et al. inoculated two types of rhizobia in a field rotation of legumes and potatoes, demonstrating a clear positive effect on potato yield with both strains. However, potato yield significantly decreased when both strains were co-inoculated as a mixture, likely due to antagonistic interactions. In the present study, L8 exhibited diverse growth-promoting functions while preventing yield reduction from antagonistic interactions (71). Furthermore, extensive research has been conducted on *B. thuringiensis* regarding its efficacy in enhancing plant stress resistance. Previous research has indicated its promising potential in the management of cotton bollworms (*Helicoverpa armigera*) and root-knot nematodes (*Meloidogyne* spp.) (72, 73). The integration of its dual capacities in promoting growth and providing protection shows significant promise for advancing sustainable agricultural practices.

## Conclusion

L8 isolates were isolated from the rhizosphere of tobacco plants and exhibited various PGP traits, such as IAA production, phosphate solubilization, and potassium solubilization. Inoculation with these multifunctional PGPR isolates improved the performance of the treated tobacco plants. Furthermore, these locally adapted multifunctional PGPR isolates can promote not only tobacco growth and quality but also the growth of various crops such as wheat, maize, and peanuts under both rotational and intercropping tillage

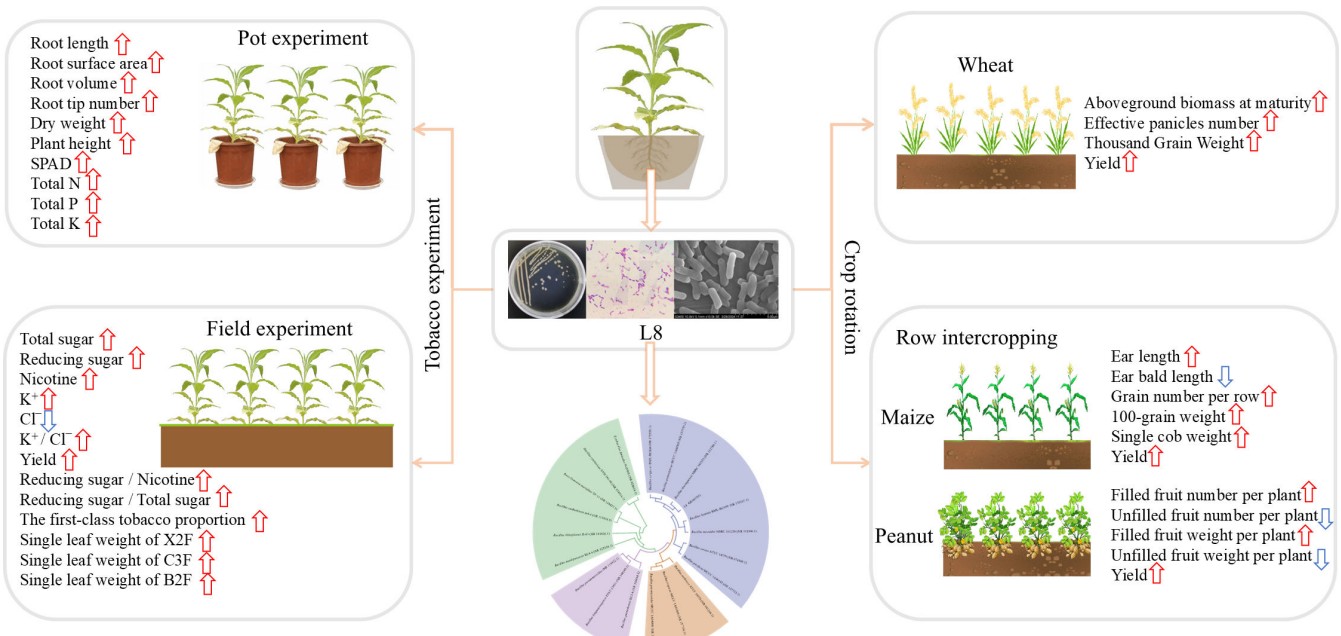

**FIG 6** Selection and application of *Bacillus thuringensis* L8 as an inoculant for plant growth promotion in tobacco and various crop rotation systems.

systems. Our findings provide a solid theoretical foundation for the deployment of L8 across diverse tillage systems in the Huang-Huai-Hai Plain, thereby advancing its potential commercialization efforts.

## ACKNOWLEDGMENTS

This work was financially supported by the Key Program for Science and Technology of CNTC (110202202030), National Key Research and Development Program of China (2024YFD1300081), the Science and Technology Program of Sichuan Provincial Branch of the CNTC (No. SCYC202306), and the Open Project of Ningbo Key Laboratory of Agricultural Germplasm Resources Mining and Environmental Regulation, College of Science and Technology, Ningbo University.

S.Z.: Data curation, Experimental operation, Writing – original draft. F.L.: Data curation, Experimental operation, Writing – original draft. L.C.: Data curation. Q.Z.: Formal analysis, Software, Visualization. Z.Z.: Formal analysis, Software, Visualization. T.W.: Formal analysis, Software, Visualization. Y.W.: Experimental operation. Z.W.: Experimental operation. Y.S.: Experimental operation. X.Y.: Supervision. W.G.: Conceptualization, Formal analysis, Supervision, Writing – review & editing. M.L.: Experimental operation. Y.W.: Data curation. Y.J.: Conceptualization, Formal analysis, Supervision, Writing – review & editing. All authors reviewed the manuscript. The author(s) read and approved the final manuscript.

## AUTHOR AFFILIATIONS

[1]Zhengzhou Tobacco Research Institute of CNTC, Zhengzhou, China
[2]College of Resources and Environment, Henan Agricultural University, Zhengzhou, China
[3]Shizhu Branch of Chongqing Tobacco Company, Shizhu, China
[4]Yuxi Branch of Yunnan Tobacco Company, Yuxi, China
[5]Tobacco Research Institute of Sichuan Provincial Tobacco Corporation, Chengdu, China
[6]Guizhou Academy of Tobacco Science, Guiyang, China
[7]Ningbo Key Laboratory of Agricultural Germplasm Resources Mining and Environmental Regulation, College of Science and Technology, Ningbo University, Ningbo, China
[8]Agricultural Technology Extension Center of Shandong Province, Jinan , China

## AUTHOR ORCIDs

Shixiang Zhang http://orcid.org/0009-0005-0835-2806
Weichang Gao http://orcid.org/0000-0001-5358-1951
Ying Jiang http://orcid.org/0000-0003-0524-1818

## FUNDING

| Funder | Grant(s) | Author(s) |
| --- | --- | --- |
| Key Program for Science and Technology of CNTC | 110202202030 | Shixiang Zhang |
| Key Program for Science and Technology of CNTC | SCYC202306 | Xingyou Yang |

## AUTHOR CONTRIBUTIONS

Shixiang Zhang, Writing – original draft, Data curation | Yulong Su, Writing – original draft | Mengsha Li, Data curation | Yue Wu, Data curation | Ying Jiang, Data curation, Writing – original draft, Writing – review and editing.

## DATA AVAILABILITY

Data will be made available on request.

## ADDITIONAL FILES

The following material is available online.

## Supplemental Material

**Supplemental material (Spectrum01879-24-s0001.docx).** Fig. S1; Table S1.

## Open Peer Review

**PEER REVIEW HISTORY (review-history.pdf).** An accounting of the reviewer comments and feedback.

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
