## [Reviewer comments · Microbiology Spectrum]

Microbiology Spectrum

Broad-spectrum applications of plant growth-promoting rhizobacteria (PGPR) across diverse crops and intricate planting systems

Shixiang Zhang, Fengmin Li, Luyin Chang, Qiyu Zhang, Zhenyu Zhang, Tianci Wu, Yanlin Wu, Zhi Wang, Yulong Su, Xingyou Yang, Weichang Gao, Mengsha Li, Yue Wu, and Ying Jiang

Corresponding Author(s): Ying Jiang, Henan Agricultural University

Review Timeline:

Submission Date:	July 28, 2024
Editorial Decision:	September 16, 2024
Revision Received:	November 18, 2024
Editorial Decision:	November 27, 2024
Revision Received:	December 4, 2024
Accepted:	December 11, 2024

Editor: Charina Gracia Banaay

Reviewer(s): Disclosure of reviewer identity is with reference to reviewer comments included in decision letter(s). The following individuals involved in review of your submission have agreed to reveal their identity: Sanghamitra Saha (Reviewer #2)

Transaction Report:

DOI: <https://doi.org/10.1128/spectrum.01879-24>

Re: Spectrum01879-24 (Broad-spectrum applications of plant growth-promoting rhizobacteria (PGPR) across diverse crops and intricate planting systems)

Dear Prof. Ying Jiang:

Thank you for the privilege of reviewing your work. Below you will find my comments, instructions from the Spectrum editorial office, and the reviewer comments.

Please go through the reviewers' comments one by one. Carefully consider each statement and make the necessary revisions to improve your paper.

In addition to what the reviewers have mentioned, especially regarding the details of the methods employed, please clearly describe the study's rationale, objectives, scope, and limitations. This should explain why reduced fertilizer application rates were not included in the treatments, as commented on by Reviewer #1. Does this relate to prevailing farmer practices and perspectives? Or is this in the light of previous studies? Is this the main problem you want to address, or is it some other issue, such as the perceived reduction in quality and yield of crops (as mentioned also in the Introduction)? This is critical since the Introduction also noted the environmental problems caused by fertilizer application, on which the study's objectives seem to hinge but were not addressed in the experimental design. Please revise the Introduction to reflect the main points you want to address and avoid confusion.

Another important consideration is including actual plant pictures showing the difference between the control and the treated plants. Pictures provide compelling visual support for the data presented.

Let me also reiterate that consistency in the terminologies (e.g., rRNA gene instead of rDNA), proper writing of scientific names (italicize all scientific names), and some language editing must be applied to improve the paper's readability and logical flow of thoughts.

Revision Guidelines

Sincerely,
Charina Gracia Banaay
Editor
Microbiology Spectrum

Reviewer #1 (Comments for the Author):

Zhang et al. investigated the effect of a rhizosphere bacterial isolate on the effect of soil nutrients and plant fitness for Tobacco cropping systems as well as for Maize-peanut crop rotation system. Authors present a very important issue of the excessive use of fertilizers and pesticides in agriculture, which indeed deteriorates soil health and poses other environmental hazards. However, the experimental design of the current study did not provide any promising solution to this problem. It is true that PGPR could help with supplemental plant nutrition, however the PGPR isolate in this study was used along with the conventional doses of fertilizers in the field experiments. What is point of using the PGPR if fertilizers dose remains the same. Yes, the application of PGPR results in higher yield, but it also means additional cost for farmers. This study would have been much more interesting and valid if the authors would have tested the effect of PGPR under the reduced dose of fertilizers. In addition, crucial details are missing in the methods section because of which it is difficult to assess the credibility of the findings. The writing needs to be improved significantly based on the scientific standards. My comments/suggestions are provided below.

L89: Please italicize the genus name here as well as at other places in the manuscript

L107-108: Please complete the sentence

L178-181: No citations are provided for the NCBI database and mega. Also, details or any reference for sequence qc should be provided here

L202-203: Was this soil sterilized?

L200-206: So, does that mean $n = 3$ or only three plants were studied for each treatment? If yes, this sample number does not seem sufficient based on the number of variables being studied. Was power analyses conducted before settling on the replicate number?

L210: Was soil sampled from each pot separately? Please provide the details of sample collection and replication.

L232-234: What is the point of using inoculants if authors are still using conventional fertilization?

L236-237: What do you mean by 'sterilized bacterial formulation'? Are you talking about the sterilized bonemeal media? Or, did you inoculate bonemeal even for the control treatment?

L239: How many plants were analyzed per treatment?

L258: 20 plants/treatment or in total? Was sampling randomized across blocks? If yes, how?

L312-314: This should be a part of methods section.

L321-322: How were reference sequences selected for this?

Fig. 3: This could go to supplemental info.

L341: There is no such thing called "164%". Please quantify it between 0-100%.

L342-344: This needs to go to methods

L344-350: No citations for any figures or tables showing this data. Please use scientific writing format instead of writing like popular press

L359-360: This sentence is redundant. The same info is being conveyed by the previous sentence

Fig 4. The caption is written very poorly. It doesn't describe what a, b, c and d parts of the figure contain. And what is difference between c and d exactly?

L500: Please don't use words like 'enormous'. Use standard scientific words.

L507: What is unstable phosphorus and potassium?

Reviewer #2 (Comments for the Author):

Title of the paper: Broad-spectrum applications of plant growth-promoting rhizobacteria (PGPR) across diverse crops and intricate planting systems.

This paper discusses the discovery bacterial isolates from around the roots of tobacco. Standard protocols like serial dilution of a soil sample were used to purify the culture. This was followed by amplification of the 16 S rRNA and one of the promising cultures was identified to be *Bacillus thuringiensis* (L8). Several tests were conducted to identify the culture; these included standard biochemical tests such as citrate utilization, methyl red test (shown in Fig 2 A and B). All pointed to the identification of a Gram-positive facultative anaerobe. For a bacterium to have growth promoting activities, tests such as IAA production, root

growth and architecture, higher levels of N, P and K should be checked. The authors did this by inoculating the strain in pots followed by transplanting tobacco seedlings in them. After thirty days of growth, they tested various parameters such as root architecture, IAA production, chlorophyll content of leaves. Field experiments were also conducted where in the bacterial culture was part of a bone meal carrier atop conventional fertilizers. They also tested whether the bacterial isolate had any effect on plant growth in wheat, maize rotation and peanut -maize intercropping.

The authors found the L8 strain to have significantly affected plant growth in pot experiments and field experiments of tobacco. Significant changes in root growth were observed as compared with the control. Inoculation of this isolate on wheat-maize rotation and peanut-maize intercropping enhanced the organic matter, available phosphorus, and potassium content in the soil and resulted in increased yields of crops.

The research findings are robust and backed by data from pot and field experiments.

Did the authors test for any pesticidal or against known tobacco pathogens? According to other research papers, Bt has known insecticidal activity and is used to control plant diseases-this would be another area that would help in decreasing usage of chemical pesticides and fertilizers.

Line 51: How does IAA secretion help in ecosystem stability? Maybe word it differently to talk about its role in plant growth

Line 324: Do the authors mean that the starting inoculum was 25 ml per 250 ml?

Line 327: Should be yeast extract instead of yeast?

Lines 344-349; Adding images of actual tobacco roots than the diagrammatic representation in Fig 7 would have been more authentic.

Line 393: What does by first-class tobacco mean?

The introduction and materials and methods could be shortened in my opinion.

Reviewer

Sanghamitra Saha

Date of submission: Sep 9th, 2024

Title of the paper: Broad-spectrum applications of plant growth-promoting rhizobacteria (PGPR) across diverse crops and intricate planting systems.

This paper discusses the discovery bacterial isolates from around the roots of tobacco. Standard protocols like serial dilution of a soil sample were used to purify the culture. This was followed by amplification of the 16 S rRNA and one of the promising cultures was identified to be *Bacillus thuringiensis* (L8). Several tests were conducted to identify the culture; these included standard biochemical tests such as citrate utilization, methyl red test (shown in Fig 2 A and B). All pointed to the identification of a Gram-positive facultative anaerobe. For a bacterium to have growth promoting activities, tests such as IAA production, root growth and architecture, higher levels of N, P and K should be checked. The authors did this by inoculating the strain in pots followed by transplanting tobacco seedlings in them. After thirty days of growth, they tested various parameters such as root architecture, IAA production, chlorophyll content of leaves. Field experiments were also conducted where in the bacterial culture was part of a bone meal carrier atop conventional fertilizers. They also tested whether the bacterial isolate had any effect on plant growth in wheat, maize rotation and peanut -maize intercropping.

The authors found the L8 strain to have significantly affected plant growth in pot experiments and field experiments of tobacco. Significant changes in root growth were observed as compared with the control. Inoculation of this isolate on wheat-maize rotation and peanut-maize intercropping enhanced the organic matter, available phosphorus, and potassium content in the soil and resulted in increased yields of crops.

The research findings are robust and backed by data from pot and field experiments.

Did the authors test for any pesticidal or against known tobacco pathogens? According to other research papers, Bt has known insecticidal activity and is used to control plant diseases-this would be another area that would help in decreasing usage of chemical pesticides and fertilizers.

Line 51: How does IAA secretion help in ecosystem stability? Maybe word it differently to talk about its role in plant growth

Line 324: Do the authors mean that the starting inoculum was 25 ml per 250 ml?

Line 327: Should be yeast extract instead of yeast?

Lines 344-349; Adding images of actual tobacco roots than the diagrammatic representation in Fig 7 would have been more authentic.

Line 393: What does by first-class tobacco mean?

The introduction and materials and methods could be shortened in my opinion.

Response to editor and reviewers

Thank you for the privilege of reviewing your work. Below you will find my comments, instructions from the Spectrum editorial office, and the reviewer comments.

Please go through the reviewers' comments one by one. Carefully consider each statement and make the necessary revisions to improve your paper.

In addition to what the reviewers have mentioned, especially regarding the details of the methods employed, please clearly describe the study's rationale, objectives, scope, and limitations. This should explain why reduced fertilizer application rates were not included in the treatments, as commented on by Reviewer #1. Does this relate to prevailing farmer practices and perspectives? Or is this in the light of previous studies? Is this the main problem you want to address, or is it some other issue, such as the perceived reduction in quality and yield of crops (as mentioned also in the Introduction)? This is critical since the Introduction also noted the environmental problems caused by fertilizer application, on which the study's objectives seem to hinge but were not addressed in the experimental design. Please revise the Introduction to reflect the main points you want to address and avoid confusion.

Another important consideration is including actual plant pictures showing the difference between the control and the treated plants. Pictures provide compelling visual support for the data presented.

Let me also reiterate that consistency in the terminologies (e.g., rRNA gene instead of rDNA), proper writing of scientific names (italicize all scientific names), and some language editing must be applied to improve the paper's readability and logical flow of thoughts.

Revision Guidelines

- Upload point-by-point responses to the issues raised by the reviewers in a file named "Response to Reviewers," NOT in your cover letter. -
- Upload a compare copy of the manuscript (without figures) as a "Marked-Up Manuscript" file.
- Upload a clean .DOC/.DOCX version of the revised manuscript and remove the previous

version.

- Each figure must be uploaded as a separate, editable, high-resolution file (TIFF or EPS preferred), and any multipanel figures must be assembled into one file.
- Any supplemental material intended for posting by ASM should be uploaded with their legends separate from the main manuscript. You can combine all supplemental material into one file (preferred) or split it into a maximum of 10 files with all associated legends included.

Sincerely,

Charina Gracia Banaay

Editor

Microbiology Spectrum

ANSWER: We deeply appreciate the suggestions and comments from editor and reviewers concerning our manuscript entitled “Broad-spectrum applications of plant growth-promoting rhizobacteria (PGPR) across diverse crops and intricate planting” [Paper # Spectrum01879-24]. Those comments are constructive and very helpful, we read through the comments carefully and made corrections to our manuscript. Based on the instructions provided in your letter, we revised the introduction and refined the methods, standardized the writing of scientific names, maintain consistency in terminology, and editorial language used to improve the readability and logical flow of ideas in the paper.

The actual potted plant images are shown below, revealing notable differences between the treated and control groups. However, due to the students’ oversight in capturing aesthetically pleasing photographs during the experiment, no actual plant pictures have been included in this manuscript. We kindly request your understanding. (In the figure, "CK" represents the control treatment, while "YC" is an abbreviation for "Yan Cao," the Chinese term for Tobacco. The identifier "YC8" was modified to "L8," incorporating the initial of the student’s name and a numerical identifier for clarity and ease of distinction in writing.)

We uploaded the files of the revised manuscript and response to all the comments. Revisions in the text are shown using tracked changes. The responses to the reviewer’s comments are marked in blue and presented following.

Reviewer #1 (Comments for the Author):

Zhang et al. investigated the effect of a rhizosphere bacterial isolate on the effect of soil nutrients and plant fitness for Tobacco cropping systems as well as for Maize-peanut crop rotation system. Authors present a very important issue of the excessive use of fertilizers and pesticides in agriculture, which indeed deteriorates soil health and poses other environmental hazards. However, the experimental design of the current study did not provide any promising solution to this problem. It is true that PGPR could help with supplemental plant nutrition, however the PGPR isolate in this study was used along with the conventional doses of fertilizers in the field experiments. What is point of using the PGPR if fertilizers dose remains the same. Yes, the application of PGPR results in higher yield, but it also means additional cost for farmers. This study would have been much more interesting and valid if the authors would have tested the effect of PGPR under the reduced dose of fertilizers. In addition, crucial details are missing in the methods section because of which it is difficult to assess the credibility of the findings. The writing needs to be improved significantly based on the scientific standards. My comments/suggestions are provided below.

ANSWER: We sincerely appreciate your valuable input. We integrate microbial inoculants with conventional chemical fertilization for several important reasons. Research has shown that microbial inoculants enhance field crops by promoting growth, improving root structure, increasing fertilizer efficiency, and boosting crop yields. These benefits complement and enhance the effects of traditional fertilization.

While reducing fertilizer use can help mitigate area-source pollution and support sustainable agriculture, completely eliminating chemical fertilizers may not be feasible or optimal. Many farmers are understandably cautious about reducing fertilizer usage, as these fertilizers reliably provide essential nutrients throughout all stages of crop growth.

Our study aimed to verify the growth-promoting effects of the PGPR strains we screened. Numerous studies have confirmed that PGPR, when used alongside conventional fertilizers, effectively promotes crop growth and enhances yield. Our findings align with these results, highlighting the potential for PGPR to improve agricultural productivity.

Medhat Rehan et al. (2023) screened several multifunctional PGPR strains in Qassim, Saudi

Arabia, namely *Streptomyces cinereoruber* strain P6-4, *Priestia megaterium* strain P12, *Rosellorea aquimaris* strain P22-2 and *Pseudomonas plecoglossicida* strain P24. The growth promotion effect of PGPR was verified in field experiments on the basis of chemical fertilizer application. The results showed that all bacterial treatments significantly increased plant growth and phosphorus absorption traits. Wang et al. (2023) inoculated *Pseudomonas moraviensis*, *Bacillus safensis* and *Falsibacillus pallidus* into wheat (*Triticum aestivum*). PGPR increased the soluble phosphorus content in soil, promoted the growth of wheat and improved the fertilizer utilization rate under the condition of conventional fertilizer. Siahaan et al. (2022) conducted a study where they applied a combination of PGPR and chemical fertilizer to tomato plants. They found that the tomato plants treated with this combination exhibited remarkable growth improvement compared to the other treatments.

Furthermore, it is worth noting that various tillage systems and soil conditions may necessitate different methods of fertilization. The objective of this study is to ascertain the efficacy of our screened strains when applied in field conditions.

We are currently engaged in further research to identify the optimal balance between microbial inoculants and chemical fertilizer application and committed to achieving sustainable agricultural production while minimizing negative impacts.

1. Medhat R, Al-Turki A, Abdelmageed AHA, Abdelhameid NM, Omar AF. 2023. Performance of Plant-Growth-Promoting Rhizobacteria (PGPR) Isolated from Sandy Soil on Growth of Tomato (*Solanum lycopersicum* L.). *Plants*. 12:1588. <https://doi.org/10.3390/plants12081588>
2. Wang Z, Zhang H, Liu L, Li SJ Xie JF, Xue X, Jiang Y. 2022. Screening of phosphate-solubilizing bacteria and their abilities of phosphorus solubilization and wheat growth promotion. *BMC Microbiol*. 22:296. <https://doi.org/10.1186/s12866-022-02715-7>
3. Siahaan P, Mangindaan NI, Singkoh MFO. 2022. Response of vegetative growth of tomato (*Solanum lycopersicum* L. VAR. MIRA) due to PGPR (plant growth-promoting rhizobacteria) combined with compost and NPK fertilizer. *Sci. Papers, Ser. Manag. Econom. Eng. Agric. Rural Dev.* (Online) 22:659-664.

-L89: Please italicize the genus name here as well as at other places in the manuscript.

ANSWER: Thank you very much for pointing this out. We have revised the non-italicized genus

name in the manuscript to ensure that they meet the format requirements.

-L107-108: Please complete the sentence.

ANSWER: We appreciate your suggestion. We have completed the sentence according to your comments.

L178-181: No citations are provided for the NCBI database and mega. Also, details or any reference for sequence qc should be provided here.

ANSWER: We appreciate your suggestion. We have added citations for the NCBI database and MEGA in this section. 16S rRNA gene sequencing of strains L8 revealed high sequence similarity with *Bacillus thuringiensis* (NR 112780). Strain L8 was submitted to the GenBank nucleotide sequence database and allocated with accession number OR545797.

L202-203: Was this soil sterilized?

ANSWER: We appreciate your comment. We did not sterilize the soil, we simply performed a treatment to remove impurities. The treatment of potted soil simulated the actual conditions in the field.

In previous reports, some research on the effects of bacteria on plant growth were evaluated under gnotobiotic microcosm conditions. Gnotobiotic microcosms are tractable, can be produced in sufficient numbers, and can be maintained under defined conditions (1). However, there is a general trade-off between gnotobiotic experiments and more complex “natural” systems. The trade-off is the high ability to detect causality in gnotobiotic experiments, yet with little relevance to the field situation, compared to the low ability to detect causality in more complex “natural” systems, but with higher relevance to field situations. It was the aim of the present experiment to investigate whether the findings are also valid under more natural conditions in a soil system.

1. Ingham RE, Trofymow J, Ingham ER., Coleman DC. 1985. Interactions of bacteria, fungi, and their nematode grazers: effects on nutrient cycling and plant growth, *Ecological Monographs* 55:119-140. <https://doi.org/10.2307/1942528>

L200-206: So, does that mean $n = 3$ or only three plants were studied for each treatment? If yes, this sample number does not seem sufficient based on the number of variables being studied. Was power analyses conducted before settling on the replicate number?

ANSWER: We appreciate your comment. The experimental treatments were conducted with three replicates. To avoid unexpected situations during the growth period, three pots of plants were cultivated for each treatment in the potting experiments. The two referenced articles (1, 2) also employed three replicates for their potting experiments. Our data results for the same treatment demonstrate consistency and significant differences compared to the control. Regarding your concern about the sample size, your feedback provides valuable insights for our future experimental designs, and we will consider increasing the sample size in our subsequent work.

1. Shameem MR, Sonali JMI, Kumar PS, Rangasamy G, Gayathri KV, Parthasarathy V. 2023. *Rhizobium mayense* sp. Nov., an efficient plant growth-promoting nitrogen-fixing bacteria isolated from rhizosphere soil. *Environmental Research* 220:115200.

<https://doi.org/10.1016/j.envres.2022.115200>

2. Wang Z, Zhang H, Liu L, Li S, Xie J, Xue X, Jiang Y. 2022. Screening of phosphate-solubilizing bacteria and their abilities of phosphorus solubilization and wheat growth promotion. *BMC microbiology* 22:296. <https://doi.org/10.1186/s12866-022-02715-7>

L210: Was soil sampled from each pot separately? Please provide the details of sample collection and replication.

ANSWER: We appreciate your comment. We have added the details of sample collection and replication. After 30 days, the plastic pot was carefully cut open to remove the soil and plant together. The plant was gently uprooted and shaken to separate the loose soil. The entire soil mixture was thoroughly mixed, and a representative sample was taken to accurately reflect the overall soil condition. This sample was then sieved through a 2 mm mesh, and a portion of the sieved sample was stored in a refrigerator at 4°C for further analysis.

-L232-234: What is the point of using inoculants if authors are still using conventional fertilization?

ANSWER: Thank you for your valuable input. While reducing fertilizer usage can decrease environmental impact and support sustainable agriculture, completely eliminating chemical fertilizers may not be feasible. Many farmers rely on them for consistent nutrient supply throughout crop growth stages. Our study aimed to validate the growth-promoting effects of the PGPR strains we selected. Also, please refer to the answer at the beginning of this response for details on this question.

L236-237: What do you mean by 'sterilized bacterial formulation'? Are you talking about the sterilized bonemeal media? Or, did you inoculate bonemeal even for the control treatment?

ANSWER: We apologize for this confusion. Sterilized bone meal served as the carrier for the bacterial strains, and the control group was also treated with sterilized bone meal. This helped to some extent in reducing growth differences in plants due to bone meal as a nutrient source. The expression in the text may be unclear enough, and we have revised it accordingly.

L239: How many plants were analyzed per treatment?

ANSWER: We appreciate your comment. At the end of the experiment, plants were harvested, a five-point sampling method was used to randomly select 10 tobacco plants from each plot.

L258: 20 plants/treatment or in total? Was sampling randomized across blocks? If yes, how?

ANSWER: We apologize for this confusion. For each plot (treatment), 20 plants were randomly selected. Sampling points were set using the five-point sampling method, with four plants randomly selected from each point, totaling 20 plants per plot. We have added details of this analysis in the relevant section of the text.

L312-314: This should be a part of methods section.

ANSWER: We appreciate your suggestion. It has been modified and incorporated into the manuscript.

L321-322: How were reference sequences selected for this?

ANSWER: We selected the 20 most homologous strains from the NCBI database, and then identified the microbial species by comparing the reference sequences of these strains. The specific steps are as follows: Data Retrieval: First, search the NCBI database for reference sequences related to the target microorganisms; Bioinformatics Analysis: Next, utilize bioinformatics tools to conduct comparative analysis of the target sequences to identify the most matching reference sequences; Species Identification: Finally, based on the comparison results, determine the species and subspecies of the microorganisms.

Fig. 3: This could go to supplemental info.

ANSWER: We appreciate your suggestion. Figure 3 has been placed in the supplemental material (Fig. S1).

L341: There is no such thing called "164%". Please quantify it between 0-100%.

ANSWER: We appreciate your suggestion. It has been modified to the correct expression and incorporated into the manuscript.

L342-344: This needs to go to methods.

ANSWER: We appreciate your suggestion. It has been modified and incorporated into the manuscript.

L344-350: No citations for any figures or tables showing this data. Please use scientific writing format instead of writing like popular press.

ANSWER: We deeply appreciate your suggestion and these comments are super useful for improving the quality of our manuscript. We have added a table reference to display this data and changed it to a more scientific writing format.

L359-360: This sentence is redundant. The same info is being conveyed by the previous sentence

ANSWER: We deeply appreciate your suggestion and these comments are useful for improving the quality of our manuscript. We have removed redundant sentences conveying the same information.

Fig 4. The caption is written very poorly. It doesn't describe what a, b, c and d parts of the figure contain. And what is difference between c and d exactly?

ANSWER: We deeply appreciate your suggestion. We have revised the title of Figure 4. Both Figure C and Figure D visualize comparative data by mapping each value in the data matrix to colors according to a defined scale. The color of each cell shows the size of the data value at the intersection of row variables and column variables. In Figure C, blue represents a negative correlation, and red represents a positive correlation. Rows represent indicators, and columns represent treatments, with each cell color reflecting the data value at the intersection of a specific indicator and treatment. Columns 1-3 represent the L8 treatment, and columns 4-5 represent the CK treatment, showing that the indicators under L8 treatment are better than those under CK treatment on the whole. In Figure D, blue represents the minimum value, red represents the maximum value, with correlations ranging from 0.2 to 1. Both rows and columns represent

indicators, displaying the correlation between the corresponding two indicators.

L500: Please don't use words like 'enormous'. Use standard scientific words.

ANSWER: We appreciate your suggestion. It has been modified and incorporated into the manuscript.

L507: What is unstable phosphorus and potassium?

ANSWER: We apologize for this confusion. There may have been a slight issue with word choice here, and we have made revisions accordingly.

Reviewer #2

-Did the authors test for any pesticidal or against known tobacco pathogens? According to other research papers, Bt has known insecticidal activity and is used to control plant diseases-this would be another area that would help in decreasing usage of chemical pesticides and fertilizers.

ANSWER: Thank you for your valuable comments, we recognize the importance of testing for pesticidal and against known tobacco pathogens, as well as exploring the insecticidal activity and application of Bt in controlling plant diseases. We will include these aspects in our future research plans and ensure that thorough investigations are carried out to address this issue.

-Line 51: How does IAA secretion help in ecosystem stability? Maybe word it differently to talk about its role in plant growth.

ANSWER: Thank you for your question. The research suggests that this strain improves nutrient utilization in the soil, enhances soil health, and plays a significant role in plant growth through the secretion of substances like auxin (IAA). We have already made revisions in the text.

-Line 324: Do the authors mean that the starting inoculum was 25 ml per 250 ml?

ANSWER: We conducted experiments to optimize the conditions for strain L8 growth and IAA production. One of the variables was the liquid volume, with different volumes of LB liquid medium placed in 250 mL flasks. The results showed that strain growth was optimal when the medium volume was 25 mL.

-Line 327: Should be yeast extract instead of yeast?

ANSWER: We appreciate your suggestion. It has been modified and incorporated into the manuscript.

-Lines 344-349: Adding images of actual tobacco roots than the diagrammatic representation in Fig 7 would have been more authentic.

ANSWER: Thank you very much for your suggestion. The pictures of the tobacco pot experiment have been placed in the above reply to the editor's comments and the reasons have been explained. The root system was processed using a root scanner, below is a screenshot of our root data and the images of tobacco roots.

SampleId	OwnerGroSpecificati	Def	File/Bg	Length(cm)	ProjArea(cm2)	SurfArea(cm2)	AvgDiam(mm)	LenPerVol(cm/m3)			
SampleId	Seedling#	NoLinks	TotalLengt	TotalProjA	TotSurfAre	AvgLength(cm)	GrpName	Backgroun	GrpName	Healthy	GrpNam
SampleId	Seedling#	Axis#	TotalLengt	TotalProjA	TotSurfAre	AvgLength(cm)	GrpName	Backgroun	GrpName	Healthy	GrpNam
CK 1	ScannerOr EPSON EP	307.4591	15.875	19.3675	160.500 2'	1	246.3635				246.3635
CK 2	ScannerOr EPSON EP	296.822	15.1765	19.558	170.490 2'	1	425.0892				425.0892
CK 3	ScannerOr EPSON EP	329.0316	16.1925	20.32	10.500 25'	1	489.8755				489.8755
CK 4	ScannerOr EPSON EP	324.6687	16.129	20.1295	50.470 25'	1	362.4574				362.4574
CK 5	ScannerOr EPSON EP	268.0237	14.605	18.3515	220.660 2'	1	282.1851				282.1851
CK 6	ScannerOr EPSON EP	267.9672	14.859	18.034	210.730 2'	1	464.1063				464.1063
L-1-1	ScannerOr EPSON EP	285.5841	16.0655	17.5895	110.680 2'	1	448.8777				448.8777
L-1-2	ScannerOr EPSON EP	355.7735	17.0815	20.828	70.460 27'	1	794.9264				794.9264
L-1-3	ScannerOr EPSON EP	354.4832	16.764	21.1455	80.410 27'	1	569.7127				569.7127
L-1-4	ScannerOr EPSON EP	349.2493	16.129	21.6535	90.190 26'	1	147.4244				147.4244
L-1-5	ScannerOr EPSON EP	287.1849	15.1765	18.923	250.520 2'	1	483.477				483.477
L-4-1	ScannerOr EPSON EP	336.7896	16.891	19.939	50.440 27'	1	591.5052				591.5052
L-4-2	ScannerOr EPSON EP	337.83	16.3195	20.701	120.360 2'	1	468.3096				468.3096
L-4-3	ScannerOr EPSON EP	283.4994	16.002	17.7165	190.840 2'	1	612.3754				612.3754
L-4-4	ScannerOr EPSON EP	286.2575	14.732	19.431	150.540 2'	1	558.5364				558.5364
L-4-5	ScannerOr EPSON EP	290.9994	15.748	18.4785	220.780 2'	1	518.5799				518.5799
L-6-1	ScannerOr EPSON EP	302.3986	16.8275	17.9705	60.890 27'	1	707.0013				707.0013
L-6-2	ScannerOr EPSON EP	276.4632	16.0655	17.2085	200.950 2'	1	373.0635				373.0635
L-6-3	ScannerOr EPSON EP	301.5599	16.3195	18.4785	110.640 2'	1	486.952				486.952
L-6-4	ScannerOr EPSON EP	316.9026	16.256	19.4945	130.630 2'	1	688.3805				688.3805
L-6-5	ScannerOr EPSON EP	291.143	14.866	19.431	160.710 2'	1	449.8232				449.8232
L-5-1	ScannerOr EPSON EP	275.5755	15.621	17.653	290.900 2'	1	402.8783				402.8783
L-5-2	ScannerOr EPSON EP	340.6445	16.764	20.32	60.280 27'	1	533.2075				533.2075
L-5-3	ScannerOr EPSON EP	148.3866	10.4775	14.1605	710.1110 1'	1	429.8043				429.8043
L-5-4	ScannerOr EPSON EP	297.4147	16.3195	18.2245	150.870 2'	1	663.9661				663.9661
L-5-5	ScannerOr EPSON EP	240.0842	14.866	16.9545	480.710 2'	1	400.9051				400.9051
L-8-1	ScannerOr EPSON EP	291.6284	16.002	18.2245	160.750 2'	1	694.2107				694.2107
L-8-2	ScannerOr EPSON EP	271.2091	14.478	17.8325	420.700 2'	1	614.0541				614.0541
L-8-3	ScannerOr EPSON EP	254.0317	14.2875	17.78	280.890 2'	1	346.7317				346.7317

-Line 393: What does by first-class tobacco mean?

ANSWER: We apologize for this confusion. “first-class tobacco” has already been modified to “high quality tobacco”.

-The introduction and materials and methods could be shortened in my opinion.

ANSWER: We appreciate your suggestion. In accordance with the reviewers' comments, we have made revisions to the Introduction and Materials and Methods sections.

Re: Spectrum01879-24R1 (Broad-spectrum applications of plant growth-promoting rhizobacteria (PGPR) across diverse crops and intricate planting systems)

Dear Prof. Ying Jiang:

Thank you for the privilege of reviewing your work. Below you will find my comments, instructions from the Spectrum editorial office, and the reviewer comments.

The manuscript has improved much from the previous version. Here are some final minor edits we suggest you implement for greater clarity:

Figure 2C - encircle L8 for easier location

Line 31 - "yet few has applied" - change to "yet a few have applied"

Line 44 - convert the percentages to fold increase (e.g. 2.5-fold) since these are beyond 100% increases in the measured parameters

Line 50 - include the specific roles determined in the study to make the statement clear - e.g. "strain L8 can play a crucial role in crop growth promotion, yield increase, and higher soil nutrient availability"

Line 52 - delete "the" in "These findings aim to the study"

Line 65 - change the article "an" to "a" - "IAA, a vital plant hormone"

Line 78 - change "had" to "have" - "these PGPRs have the potential"

Line 205 - add the word "gene" - The 16S rRNA gene sequences"

Line 213 - please clarify what the liquid volumes mean (what is the liquid you are pertaining to?). Why is there a 250 mL denominator? Is it another volume variable you tested?

Line 230 - delete "then" in the phrase "The different treatments then were..."

Line 352 - please clarify what liquid component you are referring to; what do you mean by 25 mL per 250 mL? 25 mL of what in 250 mL of what?

Line 379 - If you can, please indicate which particular variables were included in PC1 that accounted for 90.9% of variations in the clustering

Line 467 - change "conditions" to "traits"

Line 500 - change "uptake-able" to "plant-available"

Line 508 - delete "of" in "despite of"

Lines 517-518 - do not italicize "fluorescent" because it is just an adjective not part of a scientific name

Line 524 - change the word "smooth" in the phrase "this will smooth the progress" to some other more appropriate word e.g., "this will enhance the progress" or "this will promote the progress"

Line 531 - revise to "upon inoculation with these strains"

Line 533 - italicize the scientific name *Haloxylon ammodendron*

Line 562 - to clarify the statement, please revise as follows: "...likely due to antagonistic interactions. In the present study, L8 exhibited diverse growth-promoting functions..."

Line 565 - change the statement to "Previous research has indicated..."

Line 572 - change "subjected" to "treated"

Line 868 - on the Fig 2 title - revise to "...gram stain reaction, and imaging by SEM..."

Line 891 - make the Fig 6 title more descriptive so it can stand alone apart from the manuscript text, e.g., "Selection and application of *Bacillus thuringiensis* L8 as inoculant for plant growth promotion in tobacco and various crop rotation systems"

Revision Guidelines

Sincerely,
Charina Gracia Banaay
Editor
Microbiology Spectrum

Response to editor and reviewers

Thank you for the privilege of reviewing your work. Below you will find my comments, instructions from the Spectrum editorial office, and the reviewer comments.

The manuscript has improved much from the previous version. Here are some final minor edits we suggest you implement for greater clarity.

ANSWER: We deeply appreciate the suggestions made by the editor, the Spectrum editorial office and the reviewers regarding our manuscript entitled "Broad-spectrum applications of plant growth-promoting rhizobacteria (PGPR) across diverse crops and intricate planting" [Paper # Spectrum01879-24R1]. These suggestions are very helpful, and make our paper clearer and more detailed. we have carefully read the comments and made corrections to our manuscript.

We have uploaded the files of the revised manuscript and responses to all the comments. Revisions in the text are shown using tracked changes. The responses to the reviewer's comments are marked in blue and presented following.

Figure 2C - encircle L8 for easier location

ANSWER: We appreciate your suggestion. It has been modified and incorporated into the manuscript.

Line 31 - "yet few has applied" - change to "yet a few have applied"

ANSWER: We appreciate your suggestion. It has been modified and incorporated into the manuscript.

Line 44 - convert the percentages to fold increase (e.g. 2.5-fold) since these are beyond 100% increases in the measured parameters

ANSWER: We appreciate your suggestion. It has been modified and incorporated into the manuscript.

Line 50 - include the specific roles determined in the study to make the statement clear - e.g. "strain L8 can play a crucial role in crop growth promotion, yield increase, and higher soil nutrient availability"

ANSWER: We appreciate your suggestion. It has been modified and incorporated into the manuscript.

Line 52 - delete "the" in "These findings aim to the study"

ANSWER: We appreciate your suggestion. It has been modified and incorporated into the

manuscript.

Line 65 - change the article "an" to "a" - "IAA, a vital plant hormone"

ANSWER: We appreciate your suggestion. It has been modified and incorporated into the manuscript.

Line 78 - change "had" to "have" - "these PGPRs have the potential"

ANSWER: We appreciate your suggestion. It has been modified and incorporated into the manuscript.

Line 205 - add the word "gene" - "The 16S rRNA gene sequences"

ANSWER: We appreciate your suggestion. It has been modified and incorporated into the manuscript.

Line 213 - please clarify what the liquid volumes mean (what is the liquid you are pertaining to?).

Why is there a 250 mL denominator? Is it another volume variable you tested?

ANSWER: We apologize for this confusion. Here, we set different liquid volumes using LB liquid medium, with all volumes placed in 250 mL Erlenmeyer flasks. We emphasized the flask volume because, in a fixed-volume conical flask, a larger liquid volume indicates less aeration, which to some extent represents differences in aeration levels. We have made revisions accordingly in the text.

Line 230 - delete "then" in the phrase "The different treatments then were..."

ANSWER: We appreciate your suggestion. It has been modified and incorporated into the manuscript.

Line 352 - please clarify what liquid component you are referring to; what do you mean by 25 mL per 250 mL? 25 mL of what in 250 mL of what?

ANSWER: We apologize for this confusion. The phrase "25 mL per 250 mL" refers to 25 mL of LB liquid medium placed in a 250 mL conical flask. We have made corresponding revisions in the text to make the meaning clearer and more explicit.

Line 379 - If you can, please indicate which particular variables were included in PC1 that accounted for 90.9% of variations in the clustering

ANSWER: We appreciate your suggestion. The loading plot explained that 'Root Tips, RV, RV II, RSA II, RL II, Soil IAA, RSA, RL, RL I, RSA I, RV I, Root Forks, Plant N, Plant height, Plant P,

Plant K, Root AvgDiam, Soil available P, SPAD, Soil available K, Plant Fresh weight, RL IV, RSA IV, RV IV, RV V, RLV, RSA V, RV III, RSA III and RL III' contributed to PC1; 'Plant N, Plant height, Plant P, Plant K, Root AvgDiam, Soil available P, SPAD, Soil available K, Plant Fresh weight, RL IV, RSA IV, RV IV, RV V, RLV, RSA V, RV III, RSA III and RL III' contributed to both PC1 and PC2. Please understand that due to too many variables, we will not list them in the text.

Line 467 - change "conditions" to "traits"

ANSWER: We appreciate your suggestion. It has been modified and incorporated into the manuscript.

Line 500 - change "uptake-able" to "plant-available"

ANSWER: We appreciate your suggestion. It has been modified and incorporated into the manuscript.

Line 508 - delete "of" in "despite of"

ANSWER: We appreciate your suggestion. It has been modified and incorporated into the manuscript.

Lines 517-518 - do not italicize "fluorescent" because it is just an adjective not part of a scientific name

ANSWER: We appreciate your suggestion. It has been modified and incorporated into the manuscript.

Line 524 - change the word "smooth" in the phrase "this will smooth the progress" to some other more appropriate word e.g., "this will enhance the progress" or "this will promote the progress"

ANSWER: We appreciate your suggestion. It has been modified and incorporated into the manuscript.

Line 531 - revise to "upon inoculation with these strains"

ANSWER: We appreciate your suggestion. It has been modified and incorporated into the manuscript.

Line 533 - italicize the scientific name *Haloxylon ammodendron*

ANSWER: We appreciate your suggestion. It has been modified and incorporated into the manuscript.

Line 562 - to clarify the statement, please revise as follows: "...likely due to antagonistic interactions. In the present study, L8 exhibited diverse growth-promoting functions..."

ANSWER: We appreciate your suggestion. It has been modified and incorporated into the manuscript.

Line 565 - change the statement to "Previous research has indicated..."

ANSWER: We appreciate your suggestion. It has been modified and incorporated into the manuscript.

Line 572 - change "subjected" to "treated"

ANSWER: We appreciate your suggestion. It has been modified and incorporated into the manuscript.

Line 868 - on the Fig 2 title - revise to "...gram stain reaction, and imaging by SEM..."

ANSWER: We appreciate your suggestion. It has been modified and incorporated into the manuscript.

Line 891 - make the Fig 6 title more descriptive so it can stand alone apart from the manuscript text, e.g., "Selection and application of *Bacillus thuringensis* L8 as inoculant for plant growth promotion in tobacco and various crop rotation systems"

ANSWER: We appreciate your suggestion. It has been modified and incorporated into the manuscript.

Re: Spectrum01879-24R2 (Broad-spectrum applications of plant growth-promoting rhizobacteria (PGPR) across diverse crops and intricate planting systems)

Dear Prof. Ying Jiang:

Your manuscript has been accepted, and I am forwarding it to the ASM production staff for publication. Your paper will first be checked to make sure all elements meet the technical requirements. ASM staff will contact you if anything needs to be revised before copyediting and production can begin. Otherwise, you will be notified when your proofs are ready to be viewed.

Sincerely,
Charina Gracia Banaay
Editor
Microbiology Spectrum